# Simulation of Arctic snow microwave emission in surface-sensitive atmosphere channels

Melody Sandells[1], Nick Rutter[1], Kirsty Wivell[2], Richard Essery[3], Stuart Fox[2], Chawn Harlow[2], Ghislain Picard[4], Alexandre Roy[5], Alain Royer[6], and Peter Toose[7]

[1]Department of Geography and Environmental Sciences, Northumbria University, Newcastle-Upon-Tyne, UK
[2]Met Office, Exeter, UK
[3]Department of Geosciences, University of Edinburgh, Edinburgh, UK
[4]IGE, Université Grenoble Alpes, Grenoble, France
[5]Département des Sciences de l'Environnement, Université du Québec à Trois-Rivières, Trois-Rivières, Quebec, Canada
[6]Département de géomatique appliquée, Université de Sherbrooke, Sherbrooke, Canada
[7]Climate Research Division, Environment and Climate Change Canada, Toronto, Canada

**Correspondence:** Mel Sandells (melody.sandells@northumbria.ac.uk)

**Abstract.** Accurate simulations of snow emission in surface-sensitive microwave channels are needed to separate snow from atmospheric information essential for numerical weather prediction. Measurements from a field campaign in Trail Valley Creek, Inuvik, Canada during March 2018 were used to evaluate the Snow Microwave Radiative Transfer (SMRT) Model at 89 GHz and, for the first time, frequencies between 118 and 243 GHz. In situ data from 29 snow pits, including snow specific surface area, were used to calculate exponential correlation lengths to represent the snow microstructure and to initialize snowpacks for simulation with SMRT. Measured variability in snowpack properties was used to estimate uncertainty in the simulations. SMRT was coupled with the Atmospheric Radiative Transfer Simulator to account for the directionally-dependent emission and attenuation of radiation by the atmosphere. This is a major developmental step needed for top-of-atmosphere simulations of microwave brightness temperature at atmosphere-sensitive frequencies with SMRT. Nadir simulated brightness temperatures at 89, 118, 157, 183 and 243 GHz were compared with airborne measurements and with ground-based measurements at 89 GHz. Inclusion of anisotropic atmospheric radiance in SMRT had the greatest impact on brightness temperature simulations at 183 GHz and the least at 89 GHz. Medians of simulations compared well with medians of observations, with a root mean squared error of 14 K, across five frequencies and two flights (n=10). However, snowpit measurements did not capture the observed variability fully as simulations and airborne observations formed statistically different distributions. Topographical differences in simulated brightness temperature between sloped, valley and plateau areas diminished with increasing frequency as the penetration depth within the snow decreased and less emission from the underlying ground contributed to the airborne observations. Observed brightness temperature differences between flights were attributed to the deposition of a thin layer of very low density snow. This illustrates the need to account for both temporal and spatial variability in surface snow microstructure at these frequencies. Sensitivity to snow properties and the ability to reflect changes in observed brightness temperature across the frequency range for different landscapes, as demonstrated by SMRT, is a necessary condition for inclusion of atmospheric measurements at surface-sensitive frequencies in numerical weather prediction.

# 1 Introduction

Numerical weather prediction (NWP) is challenging in the Arctic due to lack of observations suitable for assimilation (Geer et al., 2014). Consequently Arctic NWP is not as accurate as for midlatitudes (Randriamampianina et al., 2021). Sparse population and extreme conditions mean that ground-based observations that could be used for assimilation are few and far-between and/or have bias in their spatial distribution (Bauer et al., 2016). In contrast, there is a wealth of satellite data at high temporal resolution at high latitudes (Lawrence et al., 2019). Atmosphere sounding data are routinely assimilated into NWP in order to initialise the forecasts. However, surface-sensitive data over Arctic regions are frequently discarded because of the difficulty in accounting for the surface component (Guedj et al., 2010; Karbou et al., 2014; Bauer et al., 2016; Hirahara et al., 2020).

Previous research has indicated benefits of the assimilation of surface-sensitive microwave data over Arctic regions, and that forecast improvements may extend to lower latitudes in the medium-range (Guedj et al., 2010; Karbou et al., 2014; Day et al., 2019), with some uncertainty in mechanisms and magnitude (Cohen et al., 2014; Overland et al., 2015). Extreme weather events in the mid-latitudes have been linked to air-mass transformation processes and Arctic amplification (Francis and Vavrus, 2012; Pithan et al., 2018; Overland et al., 2021). Mid-latitude observations have also been shown to have a strong impact on Arctic medium-range forecasts during summer (Lawrence et al., 2019). Data denial experiments within the European Centre for Medium Range Weather Forecasts NWP system highlighted the dominant impact of microwave sounding data in summer compared with winter. This was attributed in part to the reduction in number of observations used in the winter, and points to the benefits of improved methods of using these data (Lawrence et al., 2019).

Microwave observations from 19-243 GHz are sensitive to both atmosphere and surface conditions to varying degrees. Atmospheric window frequencies around 19, 37 and 89 GHz are typically chosen for applications requiring information about the surface (e.g. snow) as they are less sensitive to the atmosphere. Atmospheric sounding channels are more sensitive to the atmosphere than the surface. Frequencies around 60 and 118 GHz (oxygen absorption bands) are used to infer atmospheric temperature profile information, whereas humidity profile information is obtained from water vapour channels around 183 GHz. In the dry Arctic winter, 157 GHz can be considered a window channel. Baordo and Geer (2016) demonstrated improvements in the forecast and analysis through assimilation of humidity sounding channels (183 GHz) over snow-free land in all-sky (cloudy and clear) conditions with retrieved emissivity. A dynamic emissivity retrieval was proposed by Di Tomaso et al. (2013) and Geer et al. (2014), where land surface emissivities derived at 90 GHz were used at $183 \pm 3$ GHz and higher frequencies over snow-free land. However, this is not applicable for channels with high surface sensitivity e.g. $183 \pm 7$ GHz as the errors are too large. Following the earlier work of Bouchard et al. (2010), the relevant window channel to derive emissivity for snow- and ice-covered surfaces is 157 GHz, which is used without modification at 183 GHz. Particularly over snow, the microwave emissivity is highly spatially variable, highly dependent on frequency and has high uncertainty due to its sensitivity to the microstructure (grain size, shape and spatial arrangement at the micrometer scale) of the snow. To account for the influence of the snow on satellite atmospheric observations, the microstructure of the snow must be known well, and an accurate model of microwave scattering in snow is required to interpret the observations (Harlow and Essery, 2012; Bormann et al., 2017; Wang et al., 2017; Lawrence et al., 2019; Hirahara et al., 2020).

Numerous snow microwave scattering models have been developed with a focus on remote sensing of snow properties (e.g. Wiesmann and Mätzler, 1999; Tsang et al., 2000; Lemmetyinen et al., 2010; Ding et al., 2010; Picard et al., 2013) with no single model outperforming another (Sandells et al., 2017; Royer et al., 2017). Previous research has led to greater understanding into different microwave behaviour between these models due to relative impacts of the microstructure model,
electromagnetic model and radiative transfer solver approach (Löwe and Picard, 2015; Pan et al., 2015; Picard et al., 2018). Further understanding of model differences is facilitated through the modular structure of the Snow Microwave Radiative Transfer (SMRT) model, developed to isolate and quantify uncertainty in snow microwave scattering processes as a result of the theoretical model configuration (Picard et al., 2018). Sandells et al. (2021) evaluated SMRT against ground-based data over natural snowpacks in the 5-89 GHz range and obtained root mean squared errors of 3-12 K with Gaussian Random Field
or Teubner-Strey microstructure parameters derived from X-ray tomography and thin section images, demonstrating accuracy comparable to, or better than, other microwave scattering model evaluation studies that required optimization of the snow microstructure to obtain good agreement with observations. Through comparisons with airborne data over tundra snow at 89, 157 and 183 GHz, Harlow and Essery (2012) demonstrated a need for either surface roughness to be taken into account or a limitation placed on the microstructure-dependent scattering coefficient at these higher frequencies in order to explain
the observed emissivity spectra with the emission model used. As snow microstructure information was not available in the Harlow and Essery (2012) study, a detailed evaluation of the microwave emission model was not possible. To our knowledge, no previous studies have attempted evaluation of snow scattering models at higher frequencies useful for NWP given measured snow microstructure information.

In this study we evaluate SMRT simulated brightness temperatures (TB) against airborne data at five frequencies (89, 118,
157, 183 and 243 GHz) given in situ measured microstructure information. The purpose of this study is to demonstrate that radiative transfer simulations accounting for surface effects with SMRT can sufficiently explain the behaviour of observed airborne TB at these frequencies. This is required to improve assimilation of satellite data in numerical weather prediction but is challenging due to the spatial variability of snow at airborne measurement scales. A novel component of this is the coupling of SMRT with the Atmosphere Radiative Transfer Simulator (ARTS) (Buehler et al., 2018) to account for emission
and attenuation of the anisotropic atmospheric radiance at these higher frequencies. Data used in this study were taken as part of the MACSSIMIZE (Measurements of Arctic Clouds, Snow and Sea Ice nearby the Marginal Ice ZonE) field campaign in Trail Valley Creek (TVC), NWT, Canada in March 2018. During the campaign, multiple ground based profiles of snow specific surface area were obtained and other stratigraphic physical properties measured at multiple snow pit locations across the study area. These ground-based observations were described and analyzed in Rutter et al. (2019). Here, we use data from the 2018
field campaign to drive passive SMRT simulations at each of the snowpit locations and compare TB with limited ground-based radiometer observations at 89 GHz and with airborne TB at 89 GHz and higher frequencies. The paper is structured as follows: section 2 describes the TVC site and ground data, collection and processing of airborne data, methodology of the SMRT simulations and adjustment of TB to the level of the aircraft. SMRT simulations are compared with the ground-based radiometer observations and airborne observations in section 3, with discussion and conclusions presented in sections 4 and 5.
Access information to obtain data and code is given in section 5.

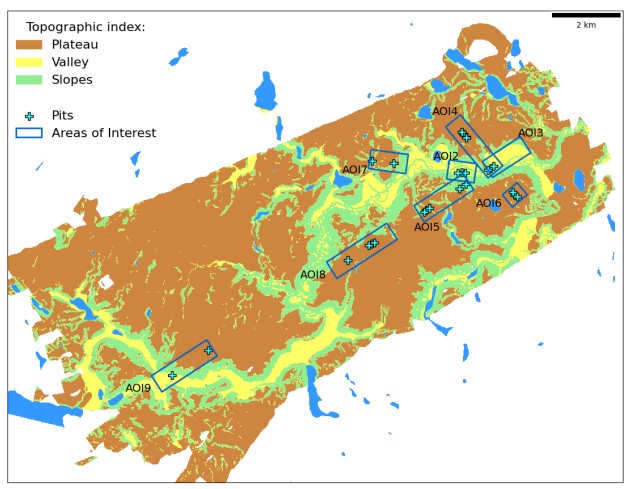

**Figure 1.** Topographic index of Trail Valley Creek, NWT, Canada with locations of snow pits and Areas Of Interest for airborne data. Adapted from Rutter et al. (2019).

## 2 Methods

### 2.1 Ground Data

Ground-based measurements of snow microstructure and microwave emission were made throughout the catchment of Trail Valley Creek (TVC), NWT, Canada (68°44'17" N 133°26'26" W) between 14-22 March 2018. The elevation range is 9 to
187 m.a.s.l and the topography is mostly gently rolling slopes with some deep valleys (Marsh et al., 2010). The dominant land surface is tussocks (37%) followed by dwarf shrubs (24%), whereas trees only constitute 2%. Further details about the vegetation characteristics are available in Grünberg et al. (2020). Figure 1 shows how the catchment was topographically divided into areas of flat upland plateau (< 5°ground slope), flat valley bottom (< 5°) and slopes (>5°) (Rutter et al., 2019) and highlights Areas of Interest (AOIs) selected for study prior to the field measurements. Further contextual information about
seasonal changes in TVC and drone-based structure-from-motion snow depth measurements within the AOIs are available in Walker et al. (2021), with some differences in AOI numbering and dimensions from this study. Snowpit measurement locations (Figure 1) were selected in order to capture a wide range of topographies, aspects and vegetation characteristics of TVC, which are also representative of the wider Arctic tundra in general. In addition, snowpit locations were linearly aligned along three flight lines to allow spatially coincident comparisons of airborne measurements with measured and simulated microwave
emissions from the surface.

Vertical profiles of snow properties (density, Specific Surface Area (SSA), temperature, stratigraphy) required to simulate microwave scattering in snow were measured in 29 snowpits. In each pit, density, SSA and temperature were measured at a 3 cm vertical resolution. Densities were measured using a $100\,\mathrm{cm}^3$ gravimetric cutter and SSA was measured using two measurement systems, an InfraRed Integrating Sphere (IRIS) (Montpetit et al., 2012) and an A2 Photonic Sensors IceCube, both of which

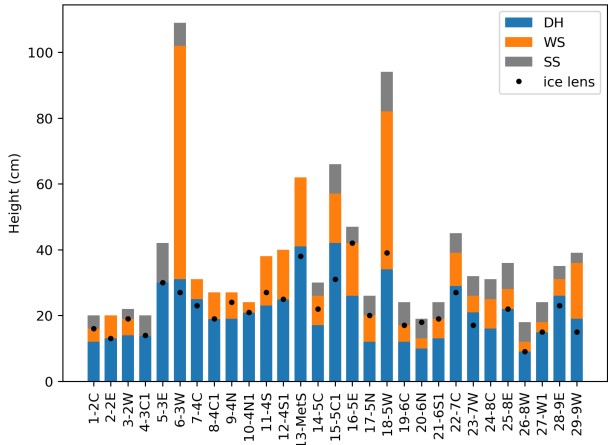

**Figure 2.** Stratigraphy of individual snowpits within Areas of Interest. Depth hoar layers (DH) are shown in blue, wind slab layers (WS) are shown in orange and surface snow (SS) is shown in grey. The location of ice lenses is shown by the black dots.

followed the method of Gallet et al. (2009) using infrared reflectance of snow samples at 1310 nm in an integrating sphere. For density and SSA, the average of two replicate samples at each position in the vertical profile were taken in the majority of snowpits in order to account for horizontal heterogeneity across the snowpit wall. Snowpack layers (including ice lenses) were identified through visual inspection and hardness tests, and classified according to Fierz et al. (2009). Additionally, following Rutter et al. (2019), snow layers were grouped into one of three microstructure types: surface snow (SS), wind slab (WS), or

depth hoar (DH), through comparative assessment of all profile measurements in combination with each other, shown in Figure 2. The majority of snowpits were between 20 and 40 cm deep. Pits 6-3W and 18-5W were located in drifts, leading to depths closer to 1 m. Depth hoar was present in all pits. Pit 5-3E did not have a wind slab layer, and only a thin wind slab layer was present in 4-3C1. Several pits did not have a fresh surface snow layer present. Almost all pits had an ice crust present, with the exception of pit 24-8C.

At ten pit locations, coincident measurements of passive microwave brightness temperature (TBs) at 89 GHz, in both vertical and horizontal polarizations, were made by a surface-based radiometer (Langlois, 2015). The radiometer was mounted on a sled at a height of approximately 1.5 m above, and at an angle nadir to near horizontal snow surfaces. A 6 dB beam width of 3° meant the measurement footprint on the snow surface was approximately 0.15 m x 0.15 m. Radiometers were calibrated using ambient (black body) and cold (liquid nitrogen) targets and had a worst case measurement error of 2 K based on 6 ambient

black body calibration checks made during the campaign. At each location, TBs measurements were made over a 6-second integration time for a minimum of three minutes. Mean TBs were calculated and the standard deviation used as a quality control flag. Three measurements were made at each site and the radiometer was moved by 2.5 m between each measurement. Coincident physical temperatures were made at both the base and within the snowpack.

| Centre frequency (GHz) | Frequency offset (GHz) | Intermediate Frequency Bandwidth (GHz) | Feature | Approximate Footprint from aircraft height ($\sim 500\,\text{m}$) |
|---|---|---|---|---|
| 88.992 | $\pm 1.075$ | 0.65 | Window | $100\,\text{m}$ |
| 118.75 | $\pm 5.0$ | 2.0 | $O_2$ | $30\,\text{m}$ |
| 157.075 | $\pm 2.6$ | 2.6 | Window | $100\,\text{m}$ |
| 183.248 | $\pm 7.0$ | 2.0 | $H_2O$ | $50\,\text{m}$ |
| 243.2 | $\pm 2.5$ | 3.0 | Window | $30\,\text{m}$ |

**Table 1.** MARSS and ISMAR channel definitions for frequencies used in this study

## 2.2 Airborne Data

During MACSSIMIZE the Facility for Airborne Atmospheric Measurements (FAAM) BAe-146 atmospheric research aircraft was based in Fairbanks, Alaska. Five flights were flown over TVC and followed a series of low-level (approximately 500m altitude) flight lines that aligned with the snowpits. This paper focuses on data for two flights, C087 and C090, on the $16^{th}$ and $20^{th}$ March 2018, as these flights were free of low cloud and occurred within the same period as ground observations were made. Airborne measurements were taken using the Microwave Airborne Radiometer Scanning System (MARSS; Mc-

Grath and Hewison, 2001) and the International Submillimetre Airborne Radiometer (ISMAR; Fox et al., 2017) on board the FAAM aircraft. Both instruments are along-track scanning radiometers containing dual-sideband heterodyne receivers measuring between 89 and 664 GHz. This paper concentrates on channels up to 243 GHz as frequencies higher than this will not have significant sensitivity to the surface except in very dry environments due to strong water vapour absorption. A summary of the channels used in this study is given in Table 1. Processing of MARSS and ISMAR data produces Rayleigh-Jeans equivalent

TBs (Fox et al., 2017).

The radiometers are mounted on the side of the aircraft, allowing both upward and downward views, and contain a rotating scan mirror with a fully configurable scan pattern. A typical scan cycle rotates through multiple upward and downward scene views, plus views of two calibration targets (one ambient and one heated). During MACSSIMIZE the instruments remained at a single downward viewing angle when over the AOIs, with calibration and zenith views in between, to increase the number

of observations taken over the surface sites. Downwelling sky observations at multiple angles are shown in Appendix A1. This paper uses observations where the instruments pointed in a near-vertical nadir direction ($\pm 5°$) when over the AOIs, which occurred during C087 and two runs of C090. Most of the MARSS and ISMAR receivers detect a single linear polarisation (of the channels studied in this paper only the 243 GHz window channel offers dual orthogonal polarisation) with the polarisation angle depending on the instrument scan angle. This must be considered for non-nadir observations; however in this paper only

near-vertical nadir observations are used where the impact of polarisation angle is minimal.

## 2.3 SMRT Modelling

The Snow Microwave Radiative Transfer (SMRT) model was previously described in Picard et al. (2018). Briefly, this is a multilayer snow scattering model suitable for passive, active and radar altimeter applications (Larue et al., 2021). It has a modular structure that allows different modelling configurations, including electromagnetic model and radiative transfer solver. For the simulations presented in this paper the Improved Born Approximation electromagnetic model and DORT radiative transfer solver were used to simulate brightness temperature emitted from the surface of the snowpack, given snowpack properties described later in this section. SMRT was coupled with ARTS to account for atmospheric emission and absorption necessary at these higher frequencies and to simulate TB at the height of the aircraft. Results presented in this paper use nadir, vertically polarized TB to evaluate SMRT against ground-based and airborne observations. Atmospheric adjustment of the ground-based radiometric data to the height of the aircraft for comparison with airborne data is described later in section 2.4.

'Base' SMRT simulations describe default parameterisations and neglect within-layer measurement variability or other potential sources of error considered later in this study. These base simulations were constructed from the three-layer dataset described in section 2.1 and shown in Figure 2. Observations of layer thickness, temperature and density were used directly to create SMRT layers. However, SMRT requires microstructure model parameters rather than the SSA observed in the field. To link with previous studies (Harlow and Essery, 2012; King et al., 2018; Vargel et al., 2020), an exponential microstructure model was chosen for this study. SSA was used to derive the required exponential correlation length with the modified Debye relationship (Mätzler, 2002; Montpetit et al., 2012):

$$l_{ex} = \alpha_{db} \frac{4(1 - \rho/\rho_i)}{SSA\rho_i} \tag{1}$$

where the Debye modification parameter $\alpha_{db}$ is assumed to be 0.75 for surface snow and wind slab layers (Mätzler, 2002) and 1.2 for depth hoar (Leinss et al., 2020) in the base simulations, $\rho$ is the snow density and $\rho_i$ is the density of pure ice. The value of $\alpha_{db}$=1.2 for depth hoar was chosen after initial assessment of the modelling strategy through a sensitivity study described below. For snowpits with dual density and SSA observation profiles, the mean layer values between profiles were used in the base simulations. Table 2 illustrates the density and SSA values used for each pit and the values taken from Rutter et al. (2019) used for missing observation values in layers that were too thin. The underlying soil surface is assumed to be flat, with a temperature of 258.15 K and permittivity of 4-0.5j based on the work of King et al. (2018) at TVC. As snowpit observations were made over an eight-day period under varying atmospheric conditions, SMRT snow layer temperatures were linearly interpolated from the air temperature at the time of the flights to the mean of the measured temperatures (263 K) in the lowest snow layer on flight days.

An ice lens was present in almost all snowpits, but occurred at different locations within the layers as shown in Figure 2. Coherent effects of ice lenses have not been implemented in SMRT, but dielectric contrast boundary effects of ice lenses are taken into account in this study. Where ice lenses were present, an additional layer was inserted into the snowpack. The recorded height of the ice lens was used to inform the strategy for amending the layering structure of the snow. As illustrated in Figure 3, for an ice lens at the boundary between layers, the thickness of the lower layer is reduced in order to maintain

**Table 2.** Snow pit properties used for base SMRT simulations. Snowpit numbering is sequential, followed by the AOI (2-9) and location within AOI (North / East / South / West or Central). Density and Specific Surface Area (SSA) are given for the Surface Snow (SS), Wind Slab (WS) and Depth Hoar (DH) layers. In layers that were too thin to measure, properties were gap-filled from the 'Missing data' values taken from Rutter et al. (2019). Flight overpass data used in this paper were from $16^{th}$ March and $20^{th}$ March 2018.

| Pit | Date | Depth [m] | Density [kg m$^{-3}$] | | | SSA [m$^2$ kg$^{-1}$] | | | Topographic Index |
|---|---|---|---|---|---|---|---|---|---|
| | | | SS | WS | DH | SS | WS | DH | |
| 1-2C | 15/03/2018 | 0.2 | - | 298 | 255 | - | 22.0 | 8.7 | Valley |
| 2-2E | 15/03/2018 | 0.2 | - | 328 | 282 | - | 30.8 | 13.5 | Valley |
| 3-2W | 14/03/2018 | 0.22 | 252 | 323 | 249 | 31.6 | 19.6 | 12.8 | Valley |
| 4-3C1 | 17/03/2018 | 0.2 | 40 | - | 230 | 31.1 | - | 13.8 | Valley |
| 5-3E | 17/03/2018 | 0.42 | 159 | - | 264 | 44.7 | - | 10.0 | Valley |
| 6-3W | 17/03/2018 | 1.09 | 132 | 368 | 270 | 43.5 | 31.0 | 13.2 | Slope |
| 7-4C | 16/03/2018 | 0.31 | - | 314 | 226 | - | 22.8 | 12.0 | Plateau |
| 8-4C1 | 16/03/2018 | 0.27 | - | 271 | 297 | - | 27.1 | 10.1 | Plateau |
| 9-4N | 16/03/2018 | 0.27 | - | 302 | 272 | - | 15.9 | 10.4 | Plateau |
| 10-4N1 | 16/03/2018 | 0.24 | - | 232 | 265 | - | 33.2 | 18.3 | Plateau |
| 11-4S | 16/03/2018 | 0.38 | - | 332 | 257 | - | 26.0 | 13.4 | Plateau |
| 12-S1 | 16/03/2018 | 0.4 | - | 308 | 262 | - | 23.8 | 13.1 | Plateau |
| 13-MetS | 22/03/2018 | 0.62 | - | 297 | 252 | - | 34.3 | 16.0 | Plateau |
| 14-5C | 21/03/2018 | 0.3 | 96 | 380 | 246 | 48.5 | 23.3 | 11.2 | Slope |
| 15-5C1 | 20/03/2018 | 0.66 | 60 | 324 | 251 | 32.3 | 24.5 | 10.6 | Plateau |
| 16-5E | 20/03/2018 | 0.47 | 65 | 310 | 257 | 41.5 | 17.4 | 12.3 | Plateau |
| 17-5N | 21/03/2018 | 0.26 | 58 | 367 | 277 | 47.6 | 20.5 | 13.0 | Slope |
| 18-5W | 20/03/2018 | 0.94 | 75 | 336 | 202 | 46.1 | 28.1 | 12.0 | Plateau |
| 19-6C | 18/03/2018 | 0.24 | 158 | 310 | 244 | 40.0 | - | 10.6 | Plateau |
| 20-6N | 18/03/2018 | 0.19 | 52 | 222 | 216 | 48.4 | 51.6 | 13.2 | Plateau |
| 21-6S1 | 18/03/2018 | 0.24 | 60 | 285 | 222 | 38.2 | 12.9 | 9.4 | Plateau |
| 22-7C | 21/03/2018 | 0.45 | 86 | 299 | 263 | 48.7 | 26.6 | 11.3 | Slope |
| 23-7W | 21/03/2018 | 0.32 | 76 | 336 | 269 | 51.9 | 34.6 | 15.3 | Plateau |
| 24-8C | 20/03/2018 | 0.31 | 90 | 287 | 238 | 48.6 | 18.8 | 12.0 | Plateau |
| 25-8E | 20/03/2018 | 0.36 | 73 | 421 | 283 | 52.8 | 28.1 | 11.7 | Plateau |
| 26-8W | 20/03/2018 | 0.18 | 94 | 250 | 196 | 51.0 | 21.9 | 8.6 | Plateau |
| 27-8W1 | 20/03/2018 | 0.24 | 80 | 205 | 258 | 56.4 | 17.8 | 9.9 | Plateau |
| 28-9E | 20/03/2018 | 0.35 | 127 | 319 | 292 | 58.4 | 22.1 | 14.6 | Plateau |
| 29-9W | 20/03/2018 | 0.39 | 38 | 307 | 349 | 88.2 | 35.6 | 14.2 | Valley |
| Missing data | - | - | 104 | 316 | 253 | 44.7 | 23.8 | 11.5 | - |

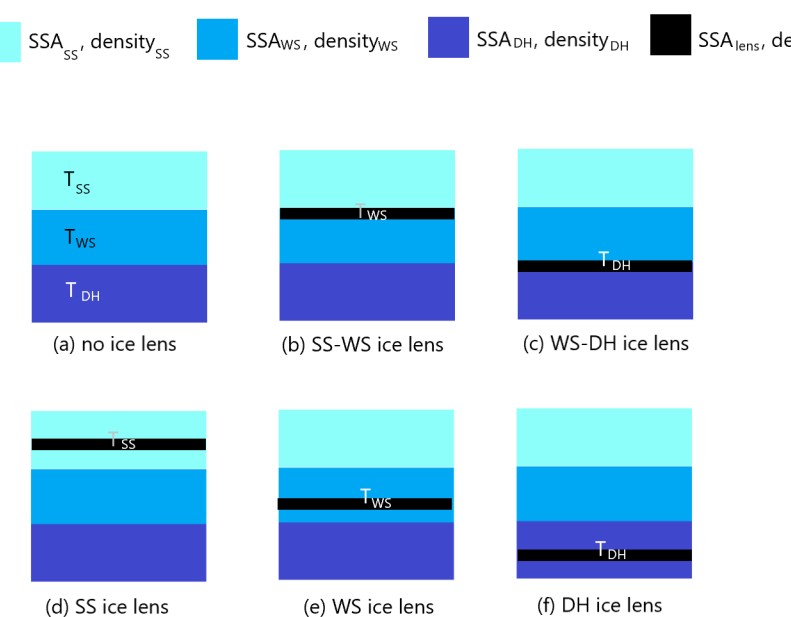

**Figure 3.** Modelling strategy to account for ice lenses in 3-layer snowpack

the correct total snow depth and the ice lens inserted, leading to a four-layer snowpack. If the ice lens occurs within a layer,
then that layer is split with the thickness of the top section given by the height of the top of the layer minus the height of the
top of the ice lens. The thickness of the lower section is recalculated to maintain total snow depth. This results in a five-layer
snowpack to represent an ice lens embedded within one of the three original layers. The ice lens density is assumed to be
$909\,\mathrm{kg\,m^{-3}}$ (Watts et al., 2016) and SSA assumed to be $100\,\mathrm{m^2\,kg^{-1}}$ (extremely weakly scattering, mainly boundary effects),
with ice lens thickness given by the field measurements. The measured ice lens thickness ranged from 1mm to 1cm, with a
mean of 2mm.

Uncertainty associated with the simulation approach was assessed using pit 9-4N as a case-study at 89 GHz. At this fre-
quency, simulations are expected to be more sensitive to processes lower in the snowpack than at other frequencies. Phenomena
observed in some pits but not accounted for in the base simulations include air gaps at the snow-soil interface and formation of
surface crusts. There is also variability in observed depth, SSA and density. Finally the modified Debye parameter $\alpha_{db}$ is not
known but often taken as 0.75 from Mätzler (2002). Leinss et al. (2020) indicated this value may be as high as 1.2 for depth
hoar, which is within the range found by Vargel et al. (2020), who considered variability in this parameter with frequency and
snow type. Here for simplicity we compare the case where all layers have $\alpha_{db}$=0.75 with the case where the depth hoar layer
has $\alpha_{db}$=1.2.

Sensitivity of simulated TB to modelling assumptions is shown in Table 3. A basal air gap was included by inserting a 5 mm
layer of low density ($10\,\mathrm{kg\,m^{-3}}$) snow and exponential correlation length of $10\,\mu\mathrm{m}$. This, however, had a negligible effect on
the TB, as did incorporating a depth observation uncertainty of 2 cm (applied to the depth hoar layer thickness). Including a

**Table 3.** Sensitivity results for snow pit 9-4N, used to define modelling protocol based on a comparison with airborne observations at 89 GHz over plateau regions of AOI4. Effect of flight C087 atmosphere (see section 2.4) included.

| Scenario | Median TB [K] | Low TB [K] | High TB [K] |
|---|---|---|---|
| a: modified Debye = 0.75 | 185.63 | | |
| b: modified depth hoar Debye = 1.2 | 179.95 | | |
| c: basal air gap | 185.49 | | |
| d: surface crust | 181.16 | | |
| e: SSA and density extremes | | 167.44 | 207.50 |
| f: depth uncertainty | | 185.79 | 185.52 |
| b + c + d + e + f | | 163.96 | 192.69 |
| b + e | | 164.60 | 198.97 |
| **AOI4 Plateau Observations** | **180.60** | **162.91** (min) **171.74** (25%ile) | **201.01** (max) **191.74** (75%ile) |

surface crust of thickness 5 mm with ice lens density and exponential correlation length of 10 microns lowered the TB by 4.5 K. A more realistic Debye modification of 1.2 applied to only the depth hoar layer resulted in a larger drop in TB of 5.7 K. This impact cannot be ignored and demonstrates a potential deficiency in the use of the 'standard' Debye correction factor of 0.75.

However, the largest impact on TB was found by representing the layer density and SSA by the largest and smallest observed values within each layer of each pit. Including all effects resulted in a TB range of 164-193 K, close to the full range of AOI4 airborne observations from the C087 flight, over areas within AOI4 classified as plateau, which was 163-201 K (see Table 3).

All simulations presented in the results section use the new Debye modification of 1.2 for the depth hoar layer (0.75 for all other layers). Surface crusts are neglected due to the difficulty in determining whether they are present or not, but could

be a source of error. Basal air gaps and uncertainty in depth are neglected due to the lack of sensitivity to them. 'Base case' simulations are driven by the median in microstructural properties, but the minimum and maximum measurements of SSA and density are also used to determine variability in simulations. Including atmospheric effects, this leads to a simulated TB range of 165-199 K for pit 9-4N (scenario b + e in Table 3), comparable to the airborne observations.

## 2.4 Adjusting for the Atmosphere

For this paper the Atmospheric Radiative Transfer Simulator (ARTS; Eriksson et al., 2011; Buehler et al., 2018) has been used to simulate the angular-dependent atmospheric radiation for SMRT. The ARTS Clear Sky (non-scattering) solver is used for a 1D atmosphere. The sensor is represented using a "top-hat" channel response in each of the two sidebands, with a frequency resolution of 0.1GHz. The simulated atmosphere accounts for the atmospheric downwelling contribution to the surface signal (radiation transmitted into the snowpack and radiation reflected by the surface) that distinguishes simulations for each flight

220 day, and is used to adjust for the layer of atmosphere between the aircraft and the surface when comparing airborne observations

to surface-based radiometer observations and simulations. Surface TB were adjusted to aircraft height using

$$T_{b,adj(\theta,\nu)} = Tr_{(\theta,\nu)} T_{b,s(\theta,\nu)} + T_{b,up(\theta,\nu)}, \tag{2}$$

where $T_{b,adj}$ is the adjusted surface TB at angle $\theta$ and frequency $\nu$, $Tr$ is the atmospheric transmission which determines the attenuation of the surface signal, $T_{b,s}$ is the unadjusted surface TB (which includes downwelling atmospheric radiation
scattered by the snow) and $T_{b,up}$ is the upwelling TB due to atmospheric emission. A flowchart illustrating the loose coupling between SMRT and ARTS and processing steps is given in Appendix A2.

The atmospheric impact is expected to be greatest for the atmospheric sounding channels due to absorption and emission by oxygen (118 GHz) and water vapour (183 GHz). However the atmospheric window channels (89, 157 and 243 GHz) also have some sensitivity to the atmosphere due to the water vapour continuum and far wings of water vapour and oxygen absorption
lines. In this paper the channels furthest from the centre of the atmospheric absorption lines at 118 and 183 GHz were chosen because strong oxygen and water vapour absorption at the channels closer to the absorption line centres mean there is little sensitivity to the surface, and these channels would be less useful for verifying SMRT.

Temperature and water vapour profiles used as input for ARTS were retrieved for each AOI in each flight. Background profiles were taken from a combination of dropsonde profiles, from sondes released before the low-level AOI runs, and profiles
from the Met Office operational global NWP model (above sonde height). The retrieval adjusts these background profiles to match aircraft-level downwelling observations in the vicinity of each AOI at $183\pm1$, $\pm3$ and $\pm7$ GHz. Because downwelling observations are only available above the aircraft, the profile below aircraft height is not adjusted in the retrieval. The height at the bottom of each profile is determined by interpolating to the mean ground height of the AOIs. Due to the instruments remaining at nadir over the AOIs, downwelling observation data at the full range of zenith viewing angles has been taken for a
period 30 seconds either side of the AOI overpass.

Within ARTS, water vapour absorption is calculated using the AER v3.6 line parameters with the MTCKD v3.2 continuum. Oxygen absorption is calculated using the Tretyakov et al. (2005) model. Simulated downwelling TBs using the ARTS absorption model configuration mentioned here are compared with observations in the Supplementary material A1 for the full range of zenith viewing angles. The figure in Appendix A1 demonstrates how atmospheric downwelling varies with viewing angle
and therefore why it is important to represent the anisotropy of the atmospheric radiance.

SMRT and therefore the ARTS configuration used return thermodynamic TBs. As stated in section 2.4, MARSS and IS-MAR processing produces Rayleigh-Jeans equivalent TBs and therefore SMRT simulations are converted to Rayleigh-Jeans equivalent before comparison with airborne observations by applying a frequency dependent offset given by $h\nu/2k$, where $h$ is Planck's constant, $\nu$ is frequency and $k$ is Boltzmann's constant. A discussion of the different TB definitions and the derivation
of the offset can be found in Han and Westwater (2000).

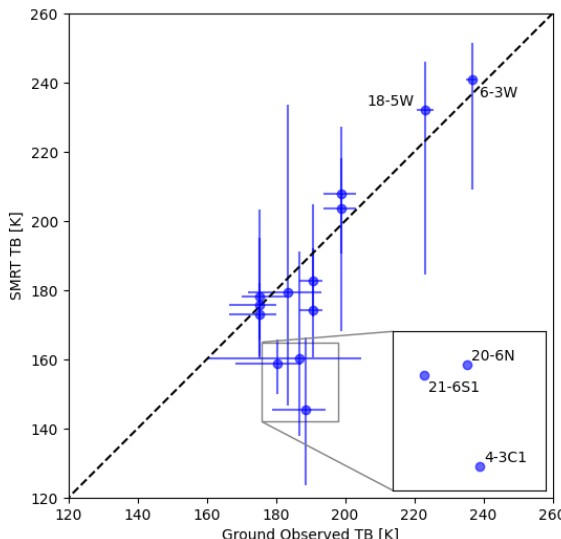

**Figure 4.** Comparison between SMRT simulations and ground-based radiometer observations at 89 GHz, nadir. Blue circles (mean of ground observations and TB simulation using mean measured snow properties), blue lines (range of ground observations and TB simulation range using combinations of maximum and minimum measured SSA and density). The zoom box is used to provide space to label these specific pits.

## 3 Results

### 3.1 SMRT Evaluation Against Ground Data

Figure 4 compares SMRT TB at 89 GHz with nadir ground-based TBs measured by the sled-mounted radiometers. SMRT simulations are the mean of the two flights, then adjusted to ground-level with the inversion of equation 2. The range of simulations capture the observations with the exception of pits 21-6S1 (plateau pit) and 4-3C1 (valley pit). The low TB simulated in 4-3C1 is later attributed to a very low surface density whereas low wind slab SSA drives the discrepancy in 21-6S1 (see section 3.2). The base simulations (shown by blue circles) tend to overestimate high TB and underestimate low TB. Overall the mean difference is -7.1 K and root mean squared difference is 16.6 K. Removal of outliers 4-3C1, 20-6N and 21-6S1 reduces the mean difference to -0.03K and root mean squared difference to 7.5K. This is quantified in terms of a difference rather than error as measurements themselves may be subject to small distortions due to shadowing of the sky and emission from the radiometers.

Ground-based radiometer observations were adjusted to the height of the aircraft and compared with airborne observations in Figure 5. Airborne observations include all those within the AOI and over the same topography classification as the pit, with the central point showing the median value and error bars indicating the interquartile range. Most observations are grouped,

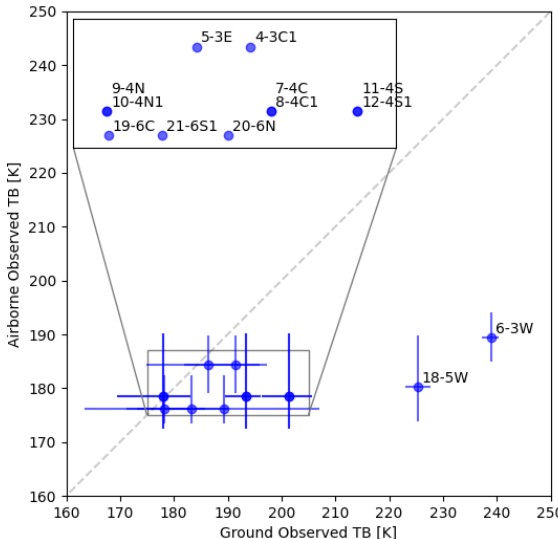

**Figure 5.** Comparison between ground-based observations of brightness temperature and airborne brightness temperature at 89 GHz for pits where observations were available. Airborne observations from both C087 and C090 flights were used, and ground-based observations have been adjusted to height of aircraft. Blue circles (mean of ground observations and median of airborne observations), blue lines (range of ground observations and inter-quartile range of airborne observations). The zoom box is used to provide space to label all pits. Note that pits 7-12 are paired pits within close proximity.

but with larger variability in the ground-based observations. Pits 6-3W (slope) and 18-5W (plateau) had a much higher TB observed on the ground than from the aircraft. These pits had the deepest snow, as shown in Figure 2, and were located in drifts. Figure 5 illustrates the challenges in using airborne data to evaluate ground-based point simulations, given that the footprint may be different in size and location.

Differences in ground vs airborne footprint location are shown in Figure 6, where data from the C087 flight have been plotted according to their calculated ground co-ordinates. Some areas of interest have pits (shown by crosses) relatively close to the line of flight e.g. AOI7, AOI9 whereas others e.g. AOI5, AOI6, AOI8 have a line of pits parallel to the flight data. TB along the airborne transects appear to show a topographic signal: plateau areas tend to have low TB and sloped or transition areas high TB. This is shown clearly in AOI7 in Figure 6, but is evident in other areas of interest. Some transects contain TB signatures not easily identifiable from the topographic map (e.g. high TB in North East of AOI8), but could be due to smaller scale heterogeneity in the underlying surface, the snow properties or vegetation. Given the difference in footprint location, it is plausible that selection of the closest airborne TB may not be representative of TB at pit locations as the underlying topography may be very different. Because of the difficulties of matching a given snowpit location with a representative airborne footprint, for comparison with SMRT simulations, all airborne observations over a particular topography class (plateau, slope, valley)

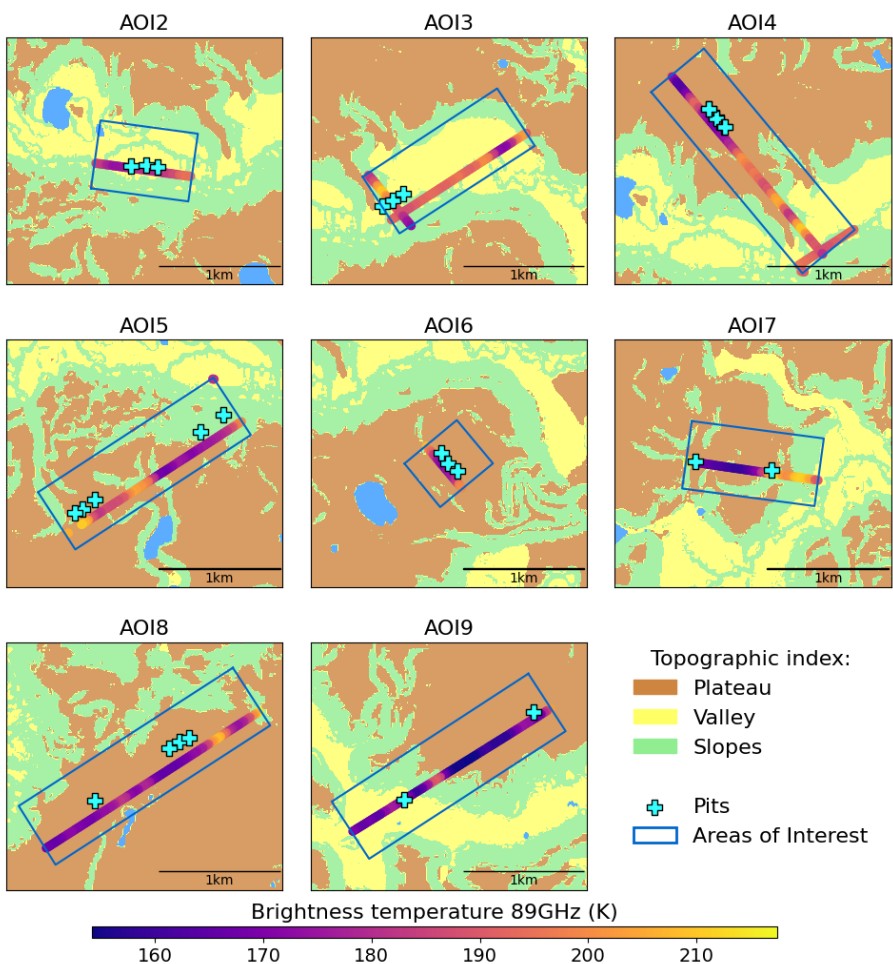

**Figure 6.** Variation in flight C087 observed airborne brightness temperature in each Area of Interest. Snow pit locations are indicated with crosses.

were grouped within each AOI. In this way, valley pit simulations were compared with all the valley airborne observations within its AOI, and likewise for the pits located on slopes and on the plateau.

## 3.2 SMRT Evaluation Against Airborne Data

Figure 7 compares the simulated TB from each of the 29 pits with the airborne observations within the same AOI and topographical index. Simulated TB at 29 pits overlapped airborne TB range in all but four pits, examined in further detail later in this section. SMRT had good agreement with ground-based TB at 6-3W but not at 4-3C1 or 21-6S1, consistent with Figure 4. Ground-based TB was not available for the Met Station snowpit. Analysis of pits grouped by their underlying topography (see Table 2) provides a test of how well SMRT simulations are able to explain the observed broad-scale spatial variability in TB.

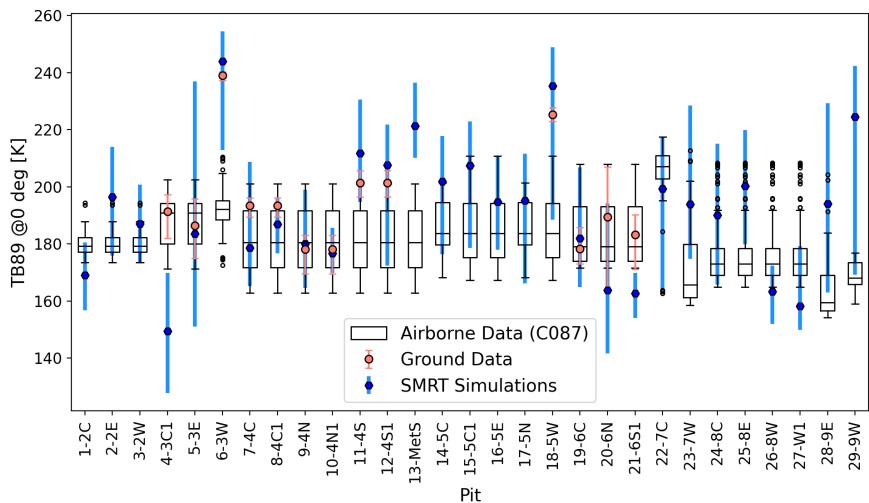

**Figure 7.** Comparison between SMRT simulations of microwave brightness temperature at 89 GHz, V-polarization, near-nadir incidence angle, ground-based measurements and flight C087 airborne observations from the MACSSIMIZE field campaign. SMRT and ground-based TB have been adjusted to the height of the aircraft. Airborne data: box (interquartile range), horizontal black line (median), vertical black lines (whiskers extending from the end of each box to 1.5 times the interquartile range), black circles (outliers beyond this range); Ground data: filled orange circle (mean), vertical orange line (range); SMRT simulations: filled blue hexagon (TB using mean measured snow properties), vertical blue line (TB range using combinations of maximum and minimum measured SSA and density).

Valley pits in AOI2 are simulated well, with overlap between simulations and observations. The western (3-2W) base simulation (blue hexagon) lies within the airborne whiskers. Variability in microstructure parameters in the Eastern and Central pits 2-2E and 1-2C leads to a larger range in simulated TB that overlaps the median of airborne TB, demonstrating that SMRT can be used to represent airborne TB adequately. Other valley pits (5-3E, 29-9W and 4-3C1) also have a large range in simulated TB.

There is close agreement between airborne median TB, ground TB and SMRT base simulation despite the large variation in microstructure at valley pit 5-3E. SMRT underestimates TB at valley pit 4-3C1. Table 2 indicates pit 4-3C1 also had an unusually low surface density. If the *missing data* value from Table 2 is used in the base simulation instead of the low surface density, TB increases from 149.3 K to 156.2 K and is therefore much closer to the observations.

Four snowpits were dug in areas classified as sloped topography. These were 6-3W, 14-5C, 17-5N and 22-7C. At 6-3W, SMRT simulations are higher than and outside of the range of airborne observations. There is, however, close agreement with ground TB measurements indicating that the airborne observations may not have observed the drift containing 6-3W. The remaining slope pits show good agreement with airborne observations, with 14-5C SMRT simulations covering the interquartile range of the airborne observations and 17-5N and 22-7C simulations covering the extent of the whiskers. For pit 22-7C, the simulations also capture the few low TB outliers.

Plateau pits are generally simulated well with the exception of the Met Station and 21-6S1. Simulated TB at the Met Station is too high compared with airborne observations, which indicates an underestimation of scattering. The Met Station is situated in AOI4 along with three sets of paired pits. Observations 7-4C and 8-4C1 were made in adjacent pits, and the ground-based radiometric observations taken at location 8-4C1 were assumed to be representative of 7-4C. Similarly the radiometric observations at 10-4N1 and 12-4S1 were assumed to be representative of 9-4N and 11-4S. The agreement between ground observations and the SMRT base case is better for the pits where the radiometric observations were made i.e. 8-4C1, 10-4N1 and 12-4S1. These adjacent pits in AOI4 give insight into the simulated microwave behaviour relative to the input data. At the central site simulated TB is lower at 7-4C than 8-4C1, which is consistent with the deeper snowpack and larger WS grains in 7-4C. The northern site is really interesting. TB at 9-4N is higher than at 10-4N1 despite smaller SSA (almost half that of 10-4N1 in both WS and DH layers). This is in contrast to the expectation that smaller SSA means larger grains, more scattering and lower TB.

The Met Station pit was the only pit dug later than flights C087 and C090 and after a strong wind event (discussed later in this section) that redistributed snow, so simulations may not be representative of the airborne observations taken beforehand. However, analysis of post-wind event flight data shows similar results to the C087 and C090 flights, suggesting this may not be the cause of the discrepancy. SSA observed at the Met Station were generally high, as shown in Table 2, but similar to pit 10-4N1. With Table 2 *missing data* SSA values applied to all layers, the base TB reduced from 221.3K to 213.0K. Conversely, pit 21-6S1 TB simulations are too low compared with both airborne and ground-based TB observations, which indicates too much scattering. Table 2 shows very low SSA for the WS layer (large grains), and values that would be more representative of depth hoar. If default *missing data* values were used for the SSA in all layers, TB increases from 162.6 K to 172.1 K, which would be closer to the observations.

Figure 8 compares SMRT simulations with observations at frequencies between 89 and 243 GHz for the two flights (C087 and C090) over all snow pits, grouped by topographic type. TB range and sensitivity of observed TB to topography decreases with increasing frequency, indicating less dependence on surface properties. Observed TB variability generally decreases from flight C087 to C090 as shown by changes in interquartile range in Figure 8. Between flights there is little change in median TB for 89 and 118 GHz, but a decrease at 157 GHz and above. SMRT simulations differ little between flights (only the atmospheric contribution changes in the simulations), leading to less overlap between simulations and observations at 183 and 243 GHz for flight C090.

Surface snowpack structure at the time of snowpit measurement may differ from the surface structure at the time of the flights. Figure 9 shows precipitation events and changes in air temperature and wind speed throughout the field campaign. Timings of three flights are also shown by the dashed vertical lines. No significant changes are expected in layer microstructure throughout the course of the field campaign as the temperature remained below freezing and only small changes in SSA can be expected over the days between flights. However, after flight C087 on $16^{th}$ March there were several snowfall events. Snow pit data from 4-3C1 on the $17^{th}$ March (Table 2) indicates the surface snow had unusually low density of 40 kg m$^{-3}$. Most snowpits after $17^{th}$ March had surface snow densities of less than 100 kg m$^{-3}$. Air temperature decreased after flight C087, with a cold spell between flight C087 and C090. Wind speed was relatively calm between flight C087 and C090 but there was

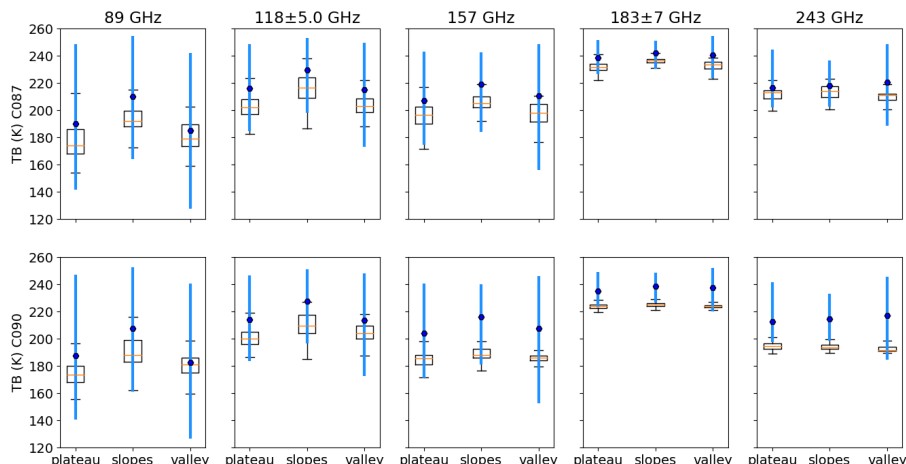

**Figure 8.** Box plot comparison between SMRT simulation (including atmosphere, adjusted to aircraft height) and airborne observations at 89, 118, 157, 183 and 243 GHz grouped by topographic type. Results for the C087 flight are shown on the top and results for the C090 flight are shown on the bottom. Airborne data: box (interquartile range), horizontal orange line (median), vertical black lines (whiskers extending from the end of each box to 1.5 times the interquartile range); SMRT simulations: filled blue hexagon (TB using mean measured snow properties), vertical blue line (TB range using combinations of maximum and minimum measured SSA and density).

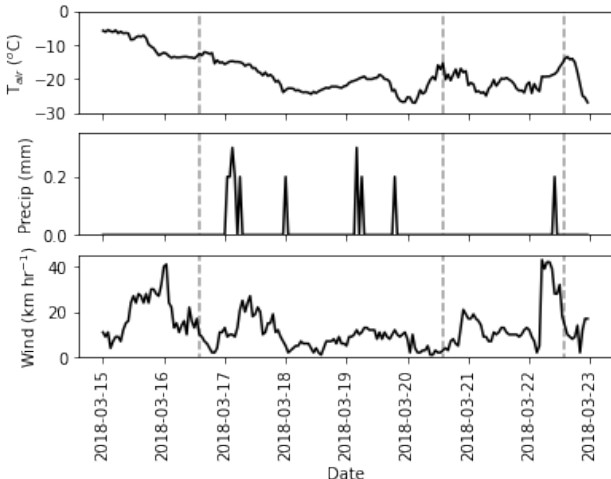

**Figure 9.** Hourly Meteorological data from Trail Valley station for duration of MACSSIMIZE campaign. Top: air temperature in degrees Celsius, Middle: Precipitation in mm, Bottom: Wind Speed in km hr$^{-1}$. Dashed lines indicate flight timings: C087 on 16$^{th}$ March, C090 on 20$^{th}$ March and C092 on 22$^{nd}$ March 2018.

**Table 4.** Effect of thin surface snow layer on simulated median brightness temperatures for different topographical land surface types (K). Brightness temperature difference is calculated for snowpits with 4-3C1 surface snow minus snowpits as measured. Negative values indicate that inclusion of low-density surface snow reduces the brightness temperature.

| Channel (GHz) | Slopes | Valley | Plateau |
|---|---|---|---|
| 89 | -0.8 | 0.6 | 0.4 |
| 118±5 | -2.2 | -0.1 | -0.4 |
| 157 | -9.5 | -7.0 | -6.2 |
| 183±7 | -3.5 | -2.8 | -2.1 |
| 243 | -13.3 | -14.4 | -12.4 |

a period of high wind speeds (maximum $43\,\mathrm{km\,hr^{-1}}$) between flight C090 on the $20^{th}$ March and flight C092 on the $22^{nd}$ March, which led to observed redistribution of surface snow after the blizzard, mostly removing snow above the ice lens in flat areas.

To examine the potential impact of surface change on TB and investigate whether this can account for the differences in observed TB between flights in Figure 8, a thin fresh surface snow layer was added to all snowpits. The additional surface snow layer was assumed to have similar properties to the surface layer of pit 4-3C1 i.e. thickness of $5\,\mathrm{cm}$, density of $40\,\mathrm{kg}$ $\mathrm{m^{-3}}$, temperature of $260\,\mathrm{K}$ and exponential correlation length of $0.1\,\mathrm{mm}$. The difference in TB is shown for each frequency in Table 4, and is shown in the Supplementary Material Figure A3. Additional surface snow decreases the brightness temperatures

at all frequencies. The absolute difference is small (<2.2 K) at 89 and 118 GHz, moderate at 183 GHz (2.1-3.5 K) and larger at 157 and 243 GHz (6.2-14.4 K). Given that the penetration depth decreases with frequency it could be expected that the effect of the surface layer should increase with frequency, but this is not the case for 183 GHz, where the effect is smaller than at 157 GHz. This suggests that emission from the atmosphere itself may dominate over the impact of the additional surface snow layer at 183 GHz, which is consistent with the higher measured and simulated emission at 183 GHz shown in Figure A1.

The importance of including the atmosphere at different frequencies is demonstrated in Figure 10. Overall, inclusion of the atmosphere reduces the root mean squared difference (RMSD) of the base simulation medians by frequency and flight from 23 K to 14 K. At an individual pit level, comparison with airborne data of the same topography classification (i.e. plateau, slopes or valleys) reveals that inclusion of the atmosphere reduces the RMSD from 35.7 K to 18.4K with the atmosphere included for flight C087 (n=145). For flight C090 the RMSD without atmosphere is 29.2K and with the atmosphere is 21.7K. The impact

of the atmosphere is largest at 183 GHz and smallest at 89 GHz. Inclusion of the atmosphere narrows the range of simulated TB. Atmospheric emission increases simulated TB, as shown by the shift in median (from blue to red dashed lines in Figure 10) despite atmospheric attenuation of emitted radiation from the snow surface. For all frequencies, median TB including the atmosphere is closer to the observations than simulations without the atmosphere. However, Kolmogorov-Smirnov 2-sample tests of distribution equivalence show that simulated distributions (either with or without atmosphere) are statistically different

to distributions of airborne observations at a 5% significance level.

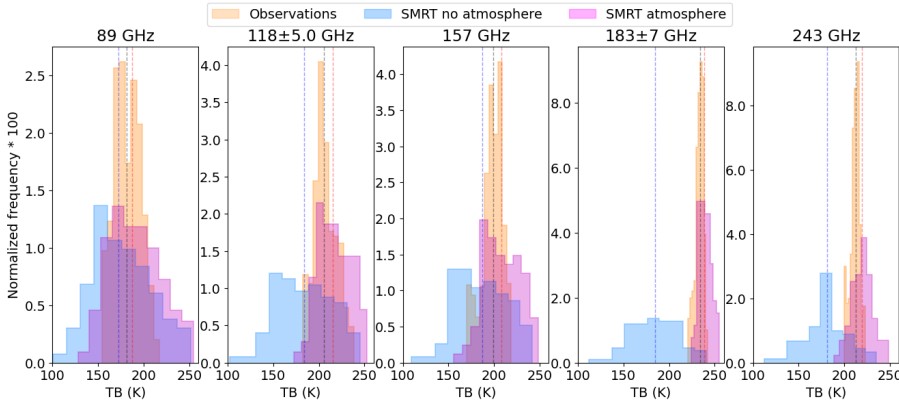

**Figure 10.** Histogram of brightness temperatures for all frequencies showing the impact of neglecting atmospheric contribution in SMRT simulations. Observations are for flight C087 only, aggregated over AOI and topographical surface type. Dashed lines show distribution medians: black for observations, blue for SMRT with no atmosphere and red for SMRT simulations incorporating atmospheric effects

## 4 Discussion

The aim of this study was to evaluate whether SMRT could be used to explain observed microwave behaviour at frequencies needed to improve numerical weather prediction in the Arctic. With anisotropic atmospheric radiance modelled with ARTS, SMRT captures the distinction between snow overlying different topography. The frequency dependence is also simulated well. The good agreement here supports the applicability of IBA electromagnetic theory at higher frequencies. With an estimated limit of wavenumber $k_0 \sim 1.5\times$ radius of spheres to keep the error of the approximation within reasonable limits, as specified by (Picard et al., 2022), the IBA upper frequency limit for the largest scattering depth hoar layer in Table 2 i.e. 8.6 m$^2$ kg$^{-1}$ is around 188 GHz. Inclusion of the atmosphere reduces simulated RMSD to a value that could be expected from comparisons with ground-based observations at frequencies more sensitive to snow. An RMSD of 14 K for the base simulations here is within the range of 13-26 K reported in the literature in the frequency range 19-89 GHz (Roy et al., 2016; Royer et al., 2017; Vargel et al., 2020) given similar in situ microstructure data.

Underlying topography is relevant at 89 GHz but becomes less relevant at higher frequencies. As the frequency increases, the penetration depth reduces and the sensor may only see the upper portion of snowpack. This is the dominant effect and results in smaller differentiation between TB classified by ground topography. However, structural changes and spatial variability in snowpack properties driven by topography may result in a topographical signal in the TB despite the signal not penetrating to the base of the snowpack. Small differences between topographical types persist even at 243 GHz in Figure 8.

Variability in ground observations of microstructure lead to a large variation in simulated TB and good overlap with airborne observations for the majority of snow pits. This demonstrates the value of making multiple measurements within the snowpack as the simulations cover a range of plausible TBs at a point given the best available snowpack structure information. Kolmagorov-Smirnov tests show that the simulations and airborne observations have different distributions even with the at-

mosphere taken into account. This may be expected as airborne observations capture more of the terrain than individual pits, which were chosen to maximise variability rather than provide random statistical sampling of the region (which would not be feasible given the number of pits required to do so). There is an issue of scale, as simulations use point measurements, whereas the airborne footprint covers an area of up to $\sim 100\,\mathrm{m}$ in diameter. Whilst snowpit simulations should lie within the range of airborne simulations and for the most part do, it is possible that the differences in ground footprint location mean that snow conditions in the snowpits were not sampled along the aircraft transects. In some cases there are clear differences in location (Figure 6) and pits 6-3W and 18-5W were located in drifts not captured in the airborne transect. The spatial extent of the drift is also smaller than the airborne footprint, so even if the flight transect had completed a direct overpass, the drift contribution to airborne observations may be limited. Further improvements could be gained with a better understanding of how to relate pit measurements to larger scale microstructure variability. This may be possible with rapid measurement instruments such as the snow micropenetrometer in conjunction with local pit calibration as demonstrated by King et al. (2020); Dutch et al. (2022).

It is crucial to know the relative thickness of layers as these can override microstructural differences by changing the penetration into lower, larger grain size layers. This is demonstrated by the paired pits in 9-4N and 10-4N1, where 9-4N TB was higher than at 10-4N1 despite smaller SSA (i.e. larger grains, more scattering) in 9-4N snowpit layers. The difference here is driven by the thinner WS layer in 10-4N1. More of the signal is proportionally affected by the DH grains than for 9-4N, leading to lower TB. The importance of the relative thickness of the depth hoar layer has already been highlighted in other studies (King et al., 2018; Rutter et al., 2019; Meloche et al., 2022) and is consistent with the higher sensitivity of surface layer changes at 94 GHz compared to lower frequencies found by Wiesmann et al. (2000).

Identification of small precipitation events with deposition of low density, small grain size surface snow will be important for use of these data in NWP. Although atmospheric conditions differ between the two flight days, the differences are too small to explain the low TB observed in flight C090. A change in microstructure rather than a change in atmospheric conditions may explain the difference in observed TB. Meteorological and in situ data presented here suggest deposition of low-density snow between the first two flights that was then removed, redistributed or was heavily compacted by wind between the second and third flights leading to similar observed TB for the first and third flights but lower TB for the middle flight. Smooth surface ice lenses facilitated wind redistribution and removal of the surface snow. Addition of a thin, low-density fresh surface snow in the simulations supports the hypothesis that the difference in observed TB is driven by snow microstructural differences between flights. In the simulations the mass of snow added is small and the exponential correlation length is also small, which means that the scattering within that layer is small. The difference in brightness temperature is likely due to the high (density-driven) dielectric contrast between layers caused by the unusually low-density fresh snow. The effect is largest at 243 GHz, where penetration depth is smallest. The difference at 183 GHz could be expected to exceed that at 157 GHz because of the shallower penetration depth at the higher frequency. It does not because the effect of the atmosphere is larger at 183 GHz.

The demonstrated ability of SMRT to represent TB variability over different snow-covered topography in TVC indicates the potential value of using SMRT to improve atmospheric retrievals given snowpack information. Vegetation may have contributed to modelling error as this was noted in many pits (see Table A1) but was not taken into account in modelling as this has yet to be implemented in SMRT. However in pits with lots of vegetation noted (shrubs to 60cm in 5-3E and 30cm shrub in pit

22-7C), SMRT base simulations are within 1.5 times the interquartile range of airborne observations. . Although contributions from twigs and grasses are likely to be small, the change in snow structure due to vegetation in pit 4-3C1 (very loose snow towards bottom, blocked by vegetation) could be a contributing factor in the discrepancy between observations and simulations. It is difficult to sample snow density and SSA within vegetation, and shrubs alter the snowpack properties, increasing depth hoar (Royer et al., 2021). In general there is good overlap with observations, with some differences between simulated TB and airborne measurements that can be explained by local variability in microstructure, changing meteorological conditions, differences in measurement location and/or footprint size.

In current numerical weather prediction models, microwave emissivity is assumed to be constant over snow-covered surfaces, is derived from a monthly climatology or is retrieved dynamically with emissivity assumed constant over frequency (Di Tomaso et al., 2013; Geer et al., 2014). However, in some channels, errors in these approaches are too large to be able to use satellite observations in the Arctic. Instead, SMRT could be used to parameterise the surface radiometric behaviour. This would require good microstructure, layer thicknesses and identification of surface snow from the NWP land surface model. Optimising assimilation of satellite observations has been identified as the most effective way to improve forecast skill in the Arctic (Laroche and Poan, 2022). NWP systems already use radiative transfer models but require higher accuracy models for snow (e.g. vertical polarisation bias at 89 GHz is currently $\sim$ -35 K: Hirahara et al., 2020, Figure 13). This should be possible with SMRT. Alternative approaches with dynamic emissivity depending on frequency can also be supported through SMRT modelling.

This modelling study encompassed dry snow conditions only, but wet snow conditions must also be considered in future work for operational numerical weather prediction models. Although the emissivity and temperature of uniformly wet snow are well-known, within-footprint spatial distribution of melt is important for simulation of brightness temperature (e.g. Vuyovich et al., 2017). The ability of land surface models to capture spatial and temporal variability in wet snow, especially freeze-thaw cycles, is important if these data are to be used to their full capacity in numerical weather prediction. Future work will focus on how we can use SMRT to quantify observation uncertainty from satellite measurements at microwave frequencies over snow-covered regions and consequently how to use the atmospheric information within them to improve weather forecasts in the Arctic.

## 5 Conclusions

In this study SMRT was evaluated at frequencies between 89 and 243 GHz in an Arctic tundra snow environment with dry snowpacks, with the atmospheric contribution estimated with ARTS. It was found that there was good agreement between simulations and airborne observations despite differences in footprint location and size. At 243 GHz, the electromagnetic model used is potentially outside the range of applicability, but the good agreement may be partly because the larger grain sizes that start to approach the wavelength of radiation are located deeper in the pack and therefore contribute less to the signal as the penetration depth decreases. Inclusion of the atmospheric emission and scattering, such as with ARTS, is essential for accurate

simulation and interpretation of ground-based, airborne and satellite observations of microwave emission at surface-sensitive atmosphere channels.

Here, a clear topography-related signal was evident at the lower frequencies, but the distinction between sloped, valley and plateau areas diminished as frequency increased. This is because the penetration depth of radiation decreases with frequency and at higher frequencies, less of the signal comes from the lower portion of the snowpack. Differences between adjacent snowpacks demonstrated that in addition to microstructure, accurate knowledge of layer thickness is also critical to determine whether the deeper snow layers are seen by the sensors. The ability of snowpack models to simulate these parameters is an
important area of research, particularly for land surface models used in Numerical Weather Prediction systems.

  Spatial variation in brightness temperatures observed with airborne instruments is reflected by the simulations, which indicates potential for use of SMRT to interpret satellite observations needed for Numerical Weather Prediction. Meteorological events i.e. the addition of fresh, low density precipitation and a later wind event that removed it over the space of a few days likely caused differences in observed brightness temperatures. The effects of this event were largest at 157 GHz and 243 GHz
as the signal is more weighted to the surface of the snow, but somewhat dampened by the atmospheric contribution at 183 GHz. This study has shown how snow microstructural and stratigraphic information can have different influences depending on the frequency of observations used. A strategy to account for both spatial and temporal variability in snow microstructure is much needed for future implementation in Numerical Weather Prediction systems, and allow use of Arctic microwave satellite observations in weather forecasts.

*Code and data availability.* Code and data to repeat these simulations are available at git@github.com:mjsandells/AESOP_paper.git. Simulations were run with SMRT commit fb330c and ARTS 2.4.0. Meteorological data for Figure 9 can be downloaded from https://climate. weather.gc.ca/historical_data/search_historic_data_e.html for station Trail Valley (WMO ID 71683), March 2018.

*Author contributions.* MS, NR, SF designed the study and wrote the AESOP proposal; MS and KW performed the SMRT simulations and data analysis; KW and SF coupled ARTS to SMRT; CH, NR, PT and RE planned and coordinated the combined airborne and ground-based
field campaign; NR, RE, AR[2] and PT made ground-based field observations; SF and CH made airborne observations; GP assisted with analysis and interpretation. All contributed to writing this paper.

*Competing interests.* At least one of the (co-)authors is a member of the editorial board of The Cryosphere.

*Acknowledgements.* This project was funded by NERC (Grant No. NE/S009280/1: Arctic Emissivity of Snow for Operational Prediction of Weather: AESOP). Data collection was made possible thanks to NERC Arctic Office UK and the Canada Arctic Partnership Bursaries

Programme (to NR and RE), Wilfrid Laurier University (Phil Marsh and Branden Walker) and Environment and Climate Change Canada. The radiometric surface-based measurements have been supported by the Natural Sciences and Engineering Research Council of Canada (NSERC) and by Polar Knowledge Canada. We thank Arvids Silis, Branden Walker and Evan Wilcox for indispensable field logistics and measurement support and Chris Derksen for help in planning the fieldwork. The MACSSIMIZE campaign was part of the Year of Polar Prediction effort, coordinated by the WMO Polar Prediction Project.

 **Appendix A:  Supplementary Figures**

**Figure A1.** Simulated (blue) vs observed mean and range (grey) downwelling brightness temperatures at the full range of zenith view angles, averaged across the AOIs at 89, 118±5.0, 157, 183±7, and 243 GHz for flight C087 (a) and C090 (b). Mean Absolute Errors are given for each channel in each flight.

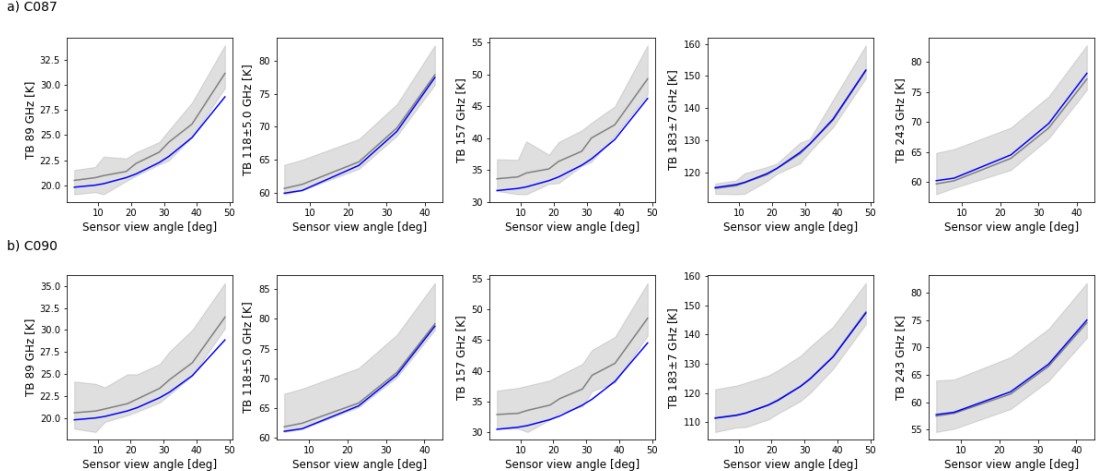

| Pit | Vegetation Notes |
|-----|------------------|
| 1-2C | Tussocks and a few shrub twigs |
| 2-2E | Tussocks and dwarf shrubs |
| 3-2W | Grass tussocks |
| 4-3C1 | Grass (very loose snow towards bottom, blocked by vegetation) |
| 5-3E | Lots of shrubs to 60cm |
| 6-3W | - |
| 7-4C | Tussocks and twigs |
| 8-4C1 | Tufts of grass |
| 9-4N | Tussocks and twigs |
| 10-4N1 | - |
| 11-4S | Tussocks and twigs |
| 12-4S1 | Lichen |
| 13-MetS | - |
| 14-5C | - |
| 15-5C1 | Lichen. Trees around pit |
| 16-5E | Further from the trees than the other. Lichen |
| 17-5N | - |
| 18-5W | Lichen, shrubs, trees around snowpit |
| 19-6C | Shrub, lichen, vegetation 7cm tall in pit |
| 20-6N | Grass and lichen |
| 21-6S1 | Lichen, small bushes |
| 22-7C | 2m shrub in area, 30cm shrub in pit |
| 23-7W | - |
| 24-8C | Lichen |
| 25-8E | - |
| 26-8W | - |
| 27-8W1 | Grass and moss |
| 28-9E | - |
| 29-9W | - |

**Table A1.** Vegetation at each pit as recorded in field notes

**Figure A2.** Flowchart demonstrating SMRT-ARTS coupling and processing steps for comparing with observations. ARTS atmospheric properties are used to calculate upwelling TB (TBup), downwelling TB (TBdown) and atmospheric transmissivity matrices used in the SMRT snowpacks. Although ARTS is configured as an initial step under the assumption of a surface blackbody, calculation of the atmospheric properties is carried out dynamically during the SMRT simulation as the matrices depend on the array of incidence angles and frequencies of the sensors. Equation 2 is used to adjust SMRT simulations to surface height for comparison with ground-based sensors (Figure 4), and for adjustment of ground observations to account for the atmosphere beneath the aircraft (Figures 5 and 7).

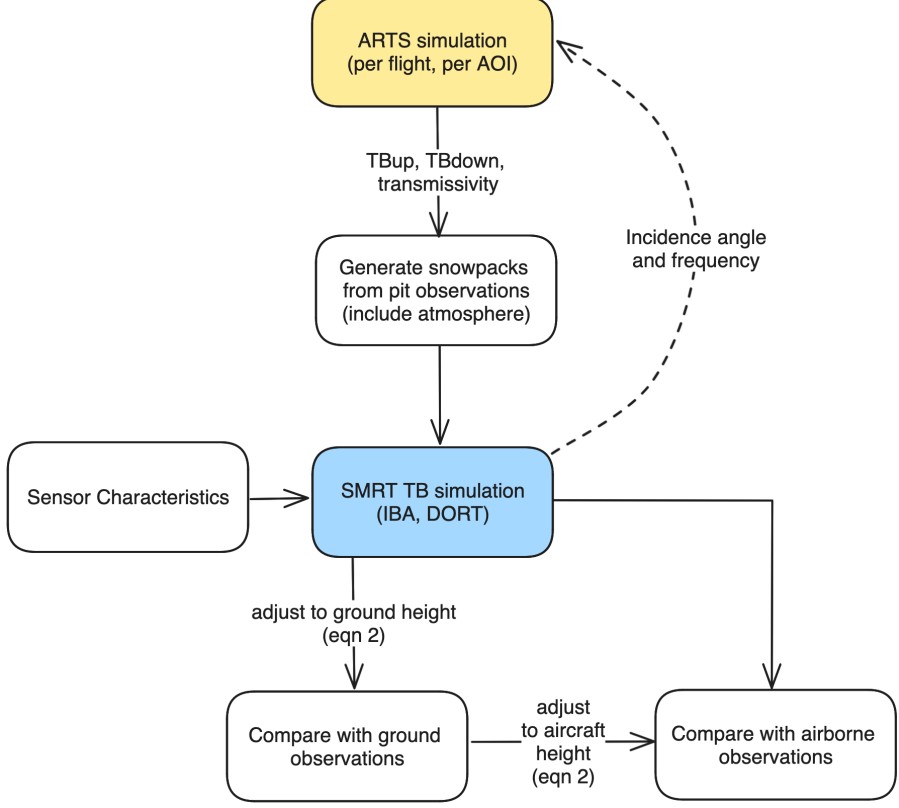

**Figure A3.** Box plot comparison between SMRT simulation (including atmosphere, adjusted to aircraft height) and airborne observations at 89, 118, 157, 183 and 243 GHz grouped by topographic type. Results for the C087 flight are shown on the top and results for the C090 flight are shown on the bottom. A thin surface snow layer has been added for flight C090 simulations. Airborne data: box (interquartile range), horizontal orange line (median), vertical black lines (whiskers extending from the end of each box to 1.5 times the interquartile range); SMRT simulations: filled blue circle (TB using mean measured snow properties), vertical blue line (TB range using combinations of maximum and minimum measured SSA and density).

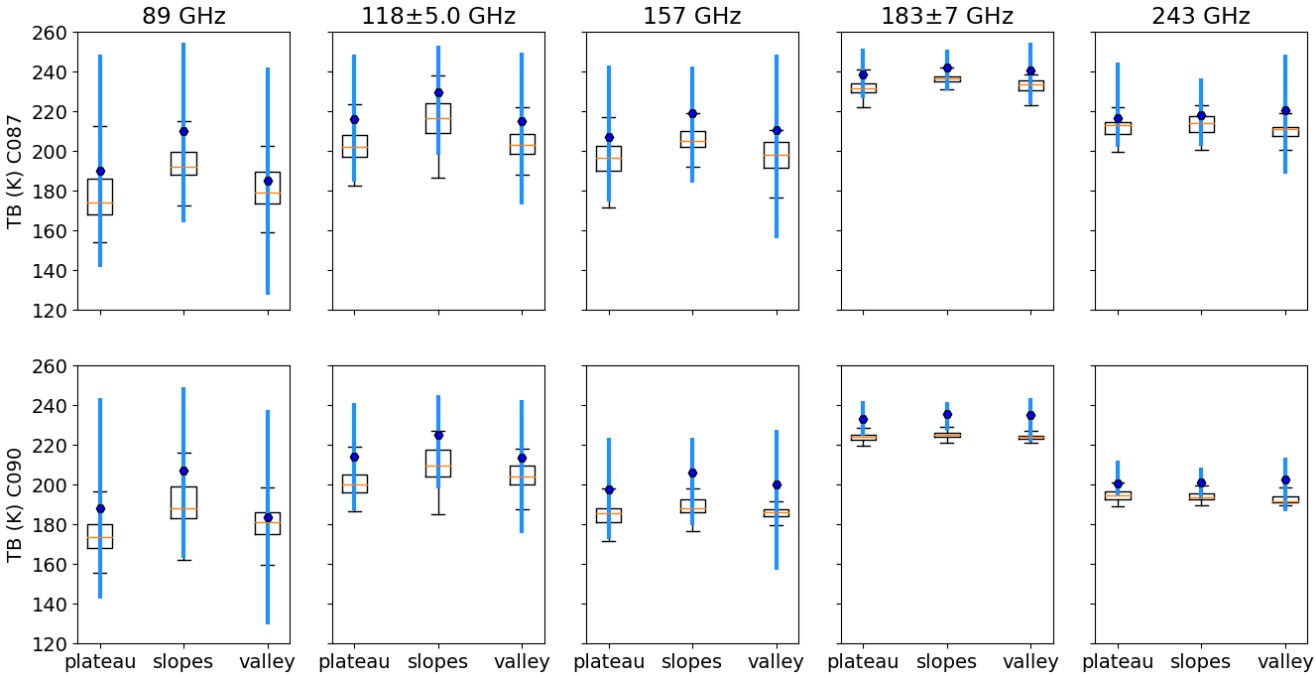

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
