# Peer review of "Simulation of Arctic snow microwave emission in surface-sensitive atmosphere channels"

_EGUsphere, 2023_

## Referee Comment (RC1)

**General comments**:

The objective of the study is not focussed on snow science, but on the search for a better use of microwave radiometer data from satellites in weather forecasting, especially in polar regions. Snow and ice surfaces produce the variable background of the atmospheric emission to be looked for. Different disciplines, dealing with atmosphere and cryosphere join here in a highly intricate way. Fortunately, the authors use up-to-date models (ARTS, SMRT) for simulating sensor signals at frequencies near 89, 118, 157, 183 and 243 GHz, i.e. at wavelengths roughly between 1 and 3 mm. Since snow-structure parameters cover a similar range, volume scattering by snow is dominant and can be highly variable. The selection of these frequencies, however, is not optimum for snow. It was based on the atmospheric properties to be sensed. The atmospheric window channels at 89, 157 and 243 GHz are highly sensitive to liquid-water clouds, to water vapour, and with increasing frequency also to ice clouds, the 118 GHz channels are used for temperature profiling around an $O_2$ line, and the 183 GHz channels are used for humidity profiling around a strong line of water vapour. Only the wing channels around these lines played a role here. In a future study, window channels at lower frequencies that are optimal for snow should be used as well. When reading the manuscript, it appears that the focus was more on snow than on the overall aspects. No information was given on the atmospheric opacity range at the given channels. Because the airborne system was optimised for the atmosphere, it is not surprising that the results were only suboptimal. Imaging microwave radiometers optimised for the surface use conical scanning with constant incidence angle. The authors found a way out of this problem by limiting the study to radiance from nadir direction. However, this limitation was a trap for various raisons as shown below.

Discrepancies between surface and aircraft observations and changes in observations between different flights were interpreted by small-scale heterogeneity and by temporal variations. Radiometric data presented were taken at an unspecified aircraft altitude. Also, data taken with a surface-based instrument on a sledge at 89 GHz were transformed to flight altitude. I cannot understand how the setup measured the snow surface in nadir direction without distortion by shadowing of the sky and by its own radiation towards the footprint. It is much better to observe at a sufficiently large nadir angle (50° to 60° off nadir, as conically scanning satellite instruments do). Then the distortion by the instrument and its setup can be negligible, and additional information by the difference between vertical and horizontal polarisation can be obtained, e.g. to separate between specular and diffuse scattering. This would help in quantifying the effect of the ice lens. Transformation to nadir direction could still be done approximately by combining SMRT and ARTS. The surface-based instrument should also be used to measure the downwelling sky radiation (tipping curves for calibration, and to determine the zenith opacity of the atmosphere). I missed information on such measurements. Indeed, atmospheric and surface radiation are linked in many ways!

**Special comments**:

1) Figure 1: I miss geographic location and altitude range. It is unclear how rugged this terrain is, how steep the slopes and therefore how large the topographic effects are.

2) Line 132: How large is the difference between the Rayleigh-Jeans equivalent TB and the physical TB based on the Planck function (especially at the highest frequency used)? Give

some typical examples. The difference between the two starts to diverge with increasing frequency and decreasing temperature.

3) The use of SSA in Table 2 and elsewhere: It would be easier for the reader to get the correlation length in mm (Eq. 1) than SSA in kg/m$^2$ because the wavelength is in mm, too.

4) The identification of snow pits in Figures 2, 4, 5, 7, and Table 2 is cumbersome when changing between text parts, Tables and Figures. Please use simple numbers from 1 to 29. You still can add things like C2, such as 1-C2 for Pit 1. This is much clearer than A0C2 because all pits are now called A0...

5) Line 179-180: What is the thickness range of the observed ice lens? How does SMRT treat its effects? Do coherent reflections between the upper and lower boundary play a role?

6) Figures 3, 8, 10: Text labels and numbers are too small.

7) Line 204 and 210 "Atmospheric correction": This term is irritating. Tb was adapted to flight height, not corrected. Only errors can be corrected in my understanding.

8) Line 232 (and elsewhere, including the abstract) "Anisotropic atmosphere": What do you mean? An atmosphere that contains anisotropic particles, such as ice crystals? Or charged particles in a magnetic field? Or do you mean anisotropic radiance? It is clear, that radiance varies with direction, even in a plane-parallel atmosphere. Therefore, the tipping-curve method has been used for a long time. But this does not mean that the atmosphere is anisotropic.

9) Line 240: "nadir ground-based TBs". See General comments, above. Measurements may be distorted (mostly enhanced) by the effects mentioned there.

10) Figure 7: I do not understand the box symbols. The key is incomplete.

11) Figure 8: Why not without atmosphere, or with a time-constant atmosphere. It is not clear if the changes are due to the atmosphere or due to the surface.

12) Figure 9 is very helpful because it shows the weather history. Its information could be used to better interpret the radiometer data. The temperature remained below freezing. No changes are expected for the ice lenses. Temperature-gradient metamorphism with slow changes only. The figure also indicates that time series of radiometric measurements at the same temporal resolution might be valuable.

13) Table 4: Text unclear. I don't see any "effect of thin...". I only see numbers. Please clarify. Why are they all negative?

14) Line 330: "This suggests that emission from the atmosphere may dominate ... at 183 GHz". It appears to me that the author did not check the actual brightness temperatures & opacities involved.

15) Figure 10 is very helpful. It is the only part where we clearly see the influence of the atmosphere. However, the analysis and description should be improved, e.g. on Line 333: "the atmosphere reduces the RMSE of the base simulation medians". I cannot see any RMSE in this figure. Do you mean the widths of the distributions shown? Please don't call this an error. And certainly not of the medians. Later, on Line 338, you mention something with respect to the distributions. I was unable do understand this text.

16) Line 345: Upper frequency limit of IBA: There is no fixed limit. The error of the approximation just increases with increasing frequency (and is larger for scattering in backward than in the forward hemisphere). "radius" should be defined, or else replaced by "correlation length".

17) Line 350: "Underlying topography": Do you really mean topography, here? Or dielectric properties of the underlying ground? The topography, in terms of slope steepness, orientation, and the solid angle of open sky above the surface point is relevant at all frequencies. The discussion that follows seems vague and not enough specific to the situations of the study.

18) The discussion on Lines 366 to 370 indicates that the selection of sensors used was not optimal. A mapping sensor with sufficient spatial resolution would have been more helpful, even if it is at a single frequency, such as 89 GHz. As an alternative a movable radiometer on a sledge would also give information on the spatial variability. Of course, this is no argument against the mentioned snow micropenetrometer. Both together would be excellent.

19) Finally, I am surprised about the large standard deviation of all simulated TB values. What is the reason? And how can you get more specific results that better focus on the actual situations?

---

## Author Comment (AC2)

Reviewer 2- Alan Geer (our responses to the comments are given in red below)

Possibly for the first time, this paper demonstrates good agreement between snow radiative transfer simulations (driven by snow pit measurements) and downlooking microwave observations at higher frequencies, i.e. 89 GHz to 243 GHz. This illustrates a path towards using snow radiative transfer, driven by multi-layer snow models, in the assimilation of microwave observations for both weather forecasting and for the inference of surface snow properties. The coupling of ARTS and SMRT models (and the demonstration of why this is important) is also an important step. The paper also illustrates some of the remaining difficulties to be solved. One of these is the occasionally large mismatches between point snow brightness temperature (TB) measurements and airborne TB measurements with fields of view up to 100m. Another is the drop in TB of up to around 20 K observed from one flight to the next, which was attributed to fresh snow on the surface, and illustrates strong temporal variability.

Overall, this paper is an important step forward, it will be of great interest to many scientists in the field, and it is well presented. However, there are a few areas where the methodology or results could be better explained, there are some possible uncertainties that might deserve more consideration, and it is important that the abstract and conclusions should clearly indicate the scope and limitations of the work.

We thank Dr. Geer for the positive overview of this paper and will bring out the scope and limitations of the work more clearly.

Main points

1) As described in the paper's abstract, coupling ARTS and SMRT is a major developmental step. However, for such an important part of the paper there is very little detail. For example it is not clear how the downwelling atmospheric radiation field is represented by ARTS and then coupled into SMRT (presumably as radiances at the quadrature angles of the discrete ordinates solver used in SMRT, but this is not stated). Assuming ARTS and SMRT are not "fully" coupled, by which I mean a discrete ordinates problem is solved simultaneously in the snow and atmosphere, I imagine that ARTS is called first to simulate the downwelling radiance field, the upwelling radiance at the observation angle, and the surface-to-aircraft transmittance. Then presumably SMRT is called and its output corrected to aircraft level with the paper's equation 2. These issues should be clearly discussed in the paper in section 2.4. It would also be good to have details of the ARTS radiative transfer solver method, mainly just to exclude the unlikely scenario that atmospheric scattering is being represented too (which could need "full" coupling of the solvers).

We appreciate and agree this needs further detail. It is correct that the two are not fully coupled and we will include a flowchart in section 2.4 to describe the steps taken. This will then sit alongside the publicly available code. We will also add the following text: 'The ARTS Clear Sky (non-scattering) solver is used for a 1D atmosphere. The sensor is represented using a "top-hat" channel response in each of the two sidebands, with a frequency resolution of 0.1GHz.'

2) Some of the descriptions of how Arctic microwave observations are used at NWP centres (with ECMWF as the main example) could be made more precise. Microwave sounding radiances are used over snow and sea-ice surfaces if the surface contribution is small enough. For example the 183+/-3 GHz channels are usually assimilated over sea-ice and snow, whereas 183+/-7 GHz channels are not. Also, one of the main problems with ice and snow surfaces, from an NWP point of view, is that a constant surface emissivity cannot be assumed. Over non-snow land surfaces, the dynamic emissivity retrieval technique typically assumes that an emissivity retrieval can be extrapolated using

a constant in frequency approximation (for example an 89 GHz retrieval is used as the surface emissivity for 183 GHz assimilation over non-snow surfaces). A few more detailed points illustrating these issues:

line 28-29: "data over Arctic regions" could more precisely be "surface-sensitive data over Arctic regions" and the reason for the data exclusion is usually the possible presence of snow and ice.

We will make this substitution

line 30: "potential benefits of .. microwave data over Arctic regions" - but some Arctic microwave data is already being assimilated operationally, particularly in summer, as illustrated in the Lawrence et al. (2019) studies, and as is described clearly on lines 35-38.

We will remove the word 'potential'

line 46: Baordo and Geer (2016) describe the assimilation of only snow-free land surface data, for the SSMIS instrument, and they eliminated surface-sensitive observations at latitudes greater than 60 degrees or for surface temperatures less than 278 K. Hence the point about using atlas in these possible-snow areas is not so relevant. Within Geer et al. (2014) there is a description of subsequent work that extended SSMIS usage over snow and sea ice surfaces following the above-described template. This actively assimilates 183+/-3 GHz and higher peaking channels. The dynamic emissivity retrieval is made at 150 GHz and then assumed to be valid also at 183 GHz, making sure the extrapolation in frequency is as small as possible (but even this small extrapolation induces errors that are too large to permit assimilation of channels that have stronger surface sensitivity, like 183+/-7 GHz). This snow and ice dynamical emissivity retrieval approach started at ECMWF even earlier with clear-sky MHS assimilation following the work of Di Tomaso et al. (2013: Assimilation of ATOVS radiances at ECMWF: third year EUMETSAT fellowship report. EUMETSAT/ECMWF Fellowship Programme Research Report No. 29, available from http://www.ecmwf.int.)

We will adapt the text to reflect this point and include the additional information on dynamic retrievals for narrow band channels, and associated references.

line 48: "microwave emissivity is highly spatially variable" - this could be a place to mention that it is also highly variable in frequency.

We will adapt the text to read 'microwave emissivity is highly spatially variable, highly dependent on frequency and has high uncertainty due to its sensitivity to the microstructure…'

Just a discussion point, but these dynamic surface approaches are continuing to be improved for NWP, and in particular we are starting to improve representations of the frequency dependence of surface emissivity. Compared to the more physical approach of the paper under review, the dynamic approach has the advantage of being able to adapt the surface to match what is in the sensor's the field of view, thus dealing with the time and space heterogeneity issues that are well illustrated in the paper, and hence they may continue to provide strong competition for the fully physical approach for some time to come.

This is a very welcome discussion point and incoming improvements to NWP. We hope that the physical approaches will underpin the representations of frequency dependence of surface emissivity and will help drive improvements in the land surface model representation also.

3) It would have been good to discuss the surface characteristics of the Trail Valley Creek site and how they relate to possible uncertainties in the surface radiative transfer. In particular, vegetation, since it appears the surface is being modelled as bare soil. The satellite pictures seem to show trees in the valleys and the possibility of grass or shrubs on the plateaus. Could vegetation have impact on the radiative transfer, particularly if it contains some liquid water, and particularly as the snow cover is not deep, e.g. 20 - 100 cm (lines 109-110)?

This is a very good point and could certainly impact the quality of the simulations. We will include this in the discussion. It's worth noting that the dominant land surface is tussocks (37%) followed by dwarf shrubs (24%), whereas trees only constitute 2% Grünberg et al., 2020 https://doi.org/10.5194/bg-17-4261-2020. Ideally we would have a radiative transfer model that simulates the effects of the vegetation, but this is not yet possible with SMRT. Vegetation was noted in many but not all pits, as shown in the Table below. We propose adding this table to the Appendix. Emission from the vegetation not accounted for could potentially contribute to an underestimation in simulated brightness temperature. However, the contributions from twigs and grasses are likely to be small. The change in snow structure due to vegetation in pit 4-3C1 (A03C1) – 'very loose snow towards bottom, blocked by vegetation' could be a contributing factor in the discrepancy between observations and simulations and will be included in the discussion.

| Pit | Vegetation notes |
|---|---|
| 1-2C | Tussocks and a few shrub twigs |
| 2-2E | Tussocks and dwarf shrubs |
| 3-2W | Grass tussocks |
| 4-3C1 | Grass (very loose snow towards bottom, blocked by vegetation) |
| 5-3E | Lots of shrubs to 60cm |
| 6-3W | - |
| 7-4C | Tussocks and twigs |
| 8-4C1 | Tufts of grass |
| 9-4N | Tussocks and twigs |
| 10-4N1 | - |
| 11-4S | Tussocks and twigs |
| 12-4S1 | Lichen |
| 13-MetS | - |
| 14-5C | - |
| 15-5C1 | Lichen. Trees around pit |
| 16-5E | Further from the trees than the other. Lichen |
| 17-5N | - |
| 18-5W | Lichen, shrubs, trees around snowpit |
| 19-6C | Shrub, lichen, vegetation 7cm tall in pit |
| 20-6N | Grass and lichen |
| 21-6S1 | Lichen, small bushes |
| 22-7C | 2m shrub in area, 30cm shrub in pit |
| 23-7W | - |
| 24-8C | Lichen |
| 25-8E | - |
| 26-8W | - |
| 27-8W1 | Grass and moss |
| 28-9E | - |
| 29-9W | - |

4) There could be some more investigation of the way the temperature profile is determined, and whether this has any bearing on the radiative transfer uncertainties. Lines 168-170 describe a linear extrapolation from the air temperature (ultimately from dropsondes?) through the snowpack to a stable lower layer temperature. Is this sufficient to represent the relative complex dependence of the snow temperature profile on the surface air temperature, particularly its insulation properties and speed of heat transfer? For example, when trying to explain the drop in brightness temperature between flights C087 and C090, could this be relevant? Looking at Figure 9, at the time of the C090 flight, could the snow still be cold after a night that dropped below -25 degrees C, and hence has not yet caught up with the rapid rise in the air temperature?

All flights were around 2pm, so residual cold from nocturnal cooling is unlikely to have persisted. The interpolation is taken from the ground station measurement, not the dropsondes (i.e. from met data in Figure 9), and we are assuming the value measured at the Met Station is representative over the whole TVC site. This is a simplification, and a better method would be e.g. snowpack modelling. However, there were only minor differences between using the measured pit temperatures and interpolated temperature estimates, so full snowpack temperature profile modelling was deemed overkill.

5) It could be worth specifying also the type of seasonal snow in the abstract and conclusions. Currently on line 406 the work is described as relating to "an Arctic tundra snow environment" but that could more precisely be "an Arctic tundra snow environment in late winter". In order to use satellite observations for weather forecasting globally and in all seasons over snow and sea-ice, we will need to be able to simulate many other snow types such as wet snow and including diurnal cycles of freeze and thaw.

This is a good point and we will make the suggested change. We will also include this in a discussion paragraph describing the limitations of this study and future research identified as a result.

Minor points

line 40 - 19, 37 and 89 GHz channels are extensively used for water vapour, cloud and precipitation assimilation, but the statement that "window frequencies around 19, 37 and 89 GHz are used to obtain information about the surface (e.g. snow)" could be misread to exclude this and to imply that these frequencies are not useful for the atmosphere.

We will replace 'used to obtain' with 'typically chosen for applications requiring'

line 44-45 - "forecast and analysis"? rather than "forecast analysis" which is confusing.

We will make this change

line 115 - if the sled measurements are nadir to a snow surface that may be sloping, is any adjustment made when sled measurements are mapped to true nadir aircraft measurements?

No adjustments were needed – sled measurements were made over near horizontal surfaces

line 196 - "representing the layer density and SSA by the largest and smallest observed values" - it's not clear whether this means within a single pit, or across all pits.

This is within each layer per single pit, and will be clarified in the text.

line 197 - the "full range of plateau airborne observations" deserves some explanation, as at this stage it's really not clear that (presumably) this means across the two flights and incorporating all plateau measurements in the relevant area illustrated in figure 6 and following the comparison strategy described in lines 262-265. It might be worth re-ordering some of this information (e.g. to put it in the section on aircraft data?)

This will be changed to 'all airborne observations from the C087 flight over areas within A04 classified as plateau'

line 221-227 - in the adjustment of the background atmospheric profile to fit aircraft-measured downwelling radiances, can the dropsonde profile below the aircraft be modified to fit observations? It's not clearly excluded in the text. And how representative is the lowermost dropsonde air temperature of the snow temperature? (See main point 4)

As suggested in the comment, because no downwelling observations were available below the aircraft, it is not possible to modify the profile below aircraft height. We can update the text to make this clearer, e.g. "Temperature and water vapour profiles used as input for ARTS were retrieved for each AOI in each flight. Background profiles were taken from a combination of dropsonde profiles, from sondes released before the low-level AOI runs, and profiles from the Met Office operational global NWP model (above sonde height). The retrieval adjusts these background profiles to match aircraft-level downwelling observations in the vicinity of each AOI at 183±1, ±3 and ±7 GHz. Because downwelling observations are only available above the aircraft, the profile below aircraft height (~590 m in the AOIs) is not adjusted in the retrieval."

For additional information, the sondes were dropped when the aircraft was at a higher altitude before the AOI runs (~7500 m in C087, and ~1800 m in C090), meaning the sonde profiles start above the altitude at which downwelling observations were made in the AOIs (~590 m). A portion of the dropsonde profile is therefore modified during the retrieval, just nothing below the height of the aircraft in the AOIs. The section of profile below the aircraft is relatively small compared to the total atmospheric profile (max altitude 80,000 m), therefore any uncertainty resulting from this is also expected to be relatively small.

The dropsonde temperatures were not used in the interpolation of snow temperatures: this information comes from the meteorological station.

Figure 4 caption (figure 5 similarly) - the significance of the square could be explained in words in the caption (the lines indicating that it is a zoom are faint and easy to miss), The caption should also explain the meaning of the error bars

In the caption we will explain that the zoom is used to provide space to label these specific outlier pits, and will make the zoom box more obvious. We will also include text to indicate the SMRT error bars arise from the variability of the simulations i.e. using the maximum and minimum density and SSA within each layer, and the observed error bars arise from the maximum and minimum of three adjacent radiometric observations.

line 241-242 - linked to Figure 4 and the need to state clearly what the error bars mean, it's not clear how the "range of simulations" mentioned here is being generated.

Please see previous response

Figure 7 and 8 captions - need a careful description of the meaning of the various boxes, whiskers and spots.

As indicated in the response to Prof. Mätzler's comment, we will include the following text: 'the airborne data box extent shows the interquartile range, the internal line represents the median and box plot whiskers extend to +/- 1.5 times the interquartile range. Open circles are outliers in the airborne observations. SMRT simulations of the base case are shown by the blue spots'

Line 383 - this RMSE calculation is a headline result from the paper, quoted in the abstract, so it should be clear how it is obtained. For me the description "RMSE of the base simulation medians by frequency and flight" is not quite clear enough. For example whether this really is the RMS of the SMRT base simulation median minus the observation median and, if I understand correctly, that means the sample over which the RMS is computed is of size ten, e.g. "across 5 frequencies and 2 flights"? Could this be more precisely described in the abstract too, noting specifically the use of medians in the calculation? Because if it is based on medians, then we might expect the RMSE comparing the errors of individual pits to individual surface categories and AOIs to be somewhat higher. That would also be a useful figure to calculate.

We will use Root Mean Square Difference rather than RMSE to describe this. This is the difference between medians so it is correct that n=10. We will clarify this in the abstract and include the individual pit RMSD for flight C087 also. For individual surface categories within AOIs against pits, the RMSD is 35.7 K excluding the atmosphere and 18.4K with the atmosphere included for flight C087 (n=145). For flight C090 the RMSD without atmosphere is 29.2K and with the atmosphere is 21.7K. These will be included in the revised manuscript, but will not be presented as a combined figure for both flights because of the difference for C090 likely caused by the thin low density snow layer.

Line 396-397 - main point 2 again: "In current numerical weather prediction models, microwave emissivity is assumed to be constant over snow-covered surfaces or derived from a monthly climatology, with errors too large to be able to use satellite observations in the Arctic": Dynamic emissivity retrievals have been used over snow and sea-ice at ECMWF to allow assimilation of 183+/-3 GHz channels (and higher peaking channels) since the work described in Di Tomaso et al. (2013) and Geer et al. (2014). Hence the snow emissivity does not usually come from atlas and it is not assumed constant in time or space (but it is assumed constant with frequency from 150 to 183 GHz). Nonetheless, the dynamic emissivity retrievals are not yet good enough to permit assimilation of strongly surface sensitive channels (e.g. 183+/-7 GHz) over snow so there still is plenty that can be improved by physical modelling as described in the paper under review.

This is a valuable discussion point – we will include these references and briefly discuss dynamic emissivity retrievals.

Line 422-423 - "the addition of fresh, low density precipitation and a later wind event that removed it over the space of a few days caused differences in observed brightness temperatures." - is the attribution of these changes in TB to the fresh snow event certain enough to be able to say for definite it "caused" it here in the conclusion, rather than to say "likely caused", for example?

While is it consistent with the simulations and could not otherwise be explained, we accept that it is not possible to attribute this conclusively and will replace 'caused' with 'likely caused' as suggested.

---

## Author Response (AR1)

Dear Patricia – thank you for your positive encouragement to make the changes to the document outlined in our previous response. Please find below a summary of the changes we have made. Reviewer comments are in black, our responses are in blue, and the *changed text in the manuscript are blue italics*. Line numbers indicating changes refer to the new version of the manuscript.

Many thanks on behalf of all authors,

Mel

Reviewer 1 - Christian Mätzler

**General comments**:
The objective of the study is not focussed on snow science, but on the search for a better use of microwave radiometer data from satellites in weather forecasting, especially in polar regions. Snow and ice surfaces produce the variable background of the atmospheric emission to be looked for. Different disciplines, dealing with atmosphere and cryosphere join here in a highly intricate way. Fortunately, the authors use up-to-date models (ARTS, SMRT) for simulating sensor signals at frequencies near 89, 118, 157, 183 and 243 GHz, i.e. at wavelengths roughly between 1 and 3 mm. Since snow-structure parameters cover a similar range, volume scattering by snow is dominant and can be highly variable. The selection of these frequencies, however, is not optimum for snow. It was based on the atmospheric properties to be sensed. The atmospheric window channels at 89, 157 and 243 GHz are highly sensitive to liquid-water clouds, to water vapour, and with increasing frequency also to ice clouds, the 118 GHz channels are used for temperature profiling around an $O_2$ line, and the 183 GHz channels are used for humidity profiling around a strong line of water vapour. Only the wing channels around these lines played a role here. In a future study, window channels at lower frequencies that are optimal for snow should be used as well. When reading the manuscript, it appears that the focus was more on snow than on the overall aspects. No information was given on the atmospheric opacity range at the given channels. Because the airborne system was optimised for the atmosphere, it is not surprising that the results were only suboptimal. Imaging microwave radiometers optimised for the surface use conical scanning with constant incidence angle. The authors found a way out of this problem by limiting the study to radiance from nadir direction. However, this limitation was a trap for various raisons as shown below.

We thank Prof. Mätzler for the comprehensive review and thoughtful comments provided on our paper.

Atmospheric opacity ranges for the different frequencies are:

|          | C087  | C090   |
|----------|-------|--------|
| 89       | 0.092 | 0.093  |
| 157      | 0.162 | 0.156  |
| 183±7    | 0.745 | 0.714  |
| 118±5.0  | 0.316 | 0.327  |
| 243      | 0.335 | 0. 321 |

These are included for the discussion record. However, the general reader may find the downwelling brightness temperature measurements from the airborne instrument more useful. These are given in the Appendix, Figure A1. This figure has been reordered according to frequency rather than instrument to be consistent with other figures). We disagree that these results are suboptimal: the challenges of point simulations vs areal observations will always exist and the purpose of this study was to demonstrate that we could explain the airborne observations through radiative transfer

*simulations, and that we can account for the surface effects in observations of the atmosphere, which we have done. The text (line 73) has been amended to 'The purpose of this study is to demonstrate that radiative transfer simulations accounting for surface effects with SMRT can sufficiently explain the behaviour of observed airborne TB at these frequencies. This is required to improve assimilation of satellite data in numerical weather prediction but is challenging due to the spatial variability of snow at airborne measurement scales.' to make the reader more aware of that we are addressing the point vs areal challenge head on.*

*The lower window frequencies that are commonly used for snow won't be very sensitive to the surface and wind-slab layers that have a big impact on the 183GHz, which is one of the key frequencies for atmospheric assimilation. For that, the 157 and 243GHz window channels are the best available. It would have been nice to have 50GHz channels to cover the key temperature sounding band, but then it would have been more critical to have e.g. 19 and 35 GHz for evaluation. To our knowledge there is no airborne system that covers the full frequency range 19-243 GHz. Conical scanning would be nice to give mapping and polarization, however, it also generally makes it very difficult/impossible to simultaneously measure the atmospheric downwelling Tb which is an important component of the validation (as demonstrated by this study). This study makes best use of a unique dataset which, to our knowledge, has no comparable equivalent elsewhere in time or location.*

Discrepancies between surface and aircraft observations and changes in observations between different flights were interpreted by small-scale heterogeneity and by temporal variations. Radiometric data presented were taken at an unspecified aircraft altitude. Also, data taken with a surface-based instrument on a sledge at 89 GHz were transformed to flight altitude. I cannot understand how the setup measured the snow surface in nadir direction without distortion by shadowing of the sky and by its own radiation towards the footprint. It is much better to observe at a sufficiently large nadir angle (50° to 60° off nadir, as conically scanning satellite instruments do). Then the distortion by the instrument and its setup can be negligible, and additional information by the difference between vertical and horizontal polarisation can be obtained, e.g. to separate between specular and diffuse scattering. This would help in quantifying the effect of the ice lens. Transformation to nadir direction could still be done approximately by combining SMRT and ARTS. The surface-based instrument should also be used to measure the downwelling sky radiation (tipping curves for calibration, and to determine the zenith opacity of the atmosphere). I missed information on such measurements. Indeed, atmospheric and surface radiation are linked in many ways!

*The aircraft altitude was approximately 500m –'(approximately 500m altitude)' has been added to the text in line 129. We have downwelling sky radiation from the aircraft measurements, as shown in Figure A1. While there may be differences between the atmospheric emission at the time of flights and the time of the ground-based measurements, these will be small at 89 GHz. The TB difference between flights was small and the variability within flights was of the order 5 K.*

*The radiometer setup is shown in the figure below:*

[Figure]

The use of a boom minimises the impact as much as possible. For the shadowing of the sky, there will be some specular component of reflection that will make the nadir measurement more sensitive to shadowing effects. However, the instrument is likely to be far from a black body and thus reflect the radiation far more than it will emit. Any metallic elements of the instrument will have reflectivities close to unity, so it may be appropriate to ignore the emission of the instrument and think of it as scattering the radiation. This means the TB measured will be larger than it would be without the instrument there, but the effect will be small. The results for the simulation of H and V shown below indicate that the error due to the radiometer set up is likely to be small.

We do have measurements at 55 deg and there are differences between H and V polarisation as shown below.

[Figure]

In terms of the simulations, we had initially looked at both sets of measurements (nadir and 55 deg) but decided to focus only on nadir measurements to simplify the paper. Nevertheless, the 55 deg simulations are interesting in themselves to look at diffuse vs specular reflection and/or the impact of ice lenses, as Prof. Mätzler indicates above.

A comparison between V-pol simulations and SBR observations at 55 deg is shown below:

[Figure]

Pit A03C1 (*now 4-3C1*) remains an outlier, but overall the simulations are not dissimilar to the nadir simulations in Figure 4, with the caveat that there are three additional pits with measurements at 55 deg: A02C (*now 1-2C)*, A02E *(2-2E)* and A02W *(3-2W)*. The mean error above is 0.4K and RMSE 12.9K, demonstrating better between agreement with observations than at nadir if all pits are included (compared with ME -7.1K and RMSE 16.6K). If pit A03C1 *(4-3C1)* is excluded, the mean error is 3.2K and RMSE 8.0K, which is less good than the nadir observations excluding the outlier pits (compared with ME -0.03K and RMSE 7.5K). Emission from the instrument may be a source of error for the nadir observations but we do not think this is a large error.

The mean difference between H and V pol measurements at 55 degrees is 22.7K, whereas for simulations (Rayleigh-Jeans, at ground level) the mean difference is 19.0K. Simulations without ice lenses result in a TB difference of 11.7K. We consider SMRT performs reasonably well at both nadir and at 55 degrees with ice lenses present.

*(For discussion record – the manuscript has not been changed in this regard)*

**Special comments**:
1) Figure 1: I miss geographic location and altitude range. It is unclear how rugged this terrain is, how steep the slopes and therefore how large the topographic effects are.

The details of the research basin will be added to the manuscript. Data were collected within the within the research basin of Trail Valley Creek (TVC), NWT, Canada (68◦44' N, 133◦33' W). The elevation range is 9 to 187 m.a.s.l and the topography is mostly gently rolling slopes with some deep

valleys (Marsh et al., 2010 https://doi.org/10.1002/hyp.7786). For further details about the vegetation characteristics, see Grünberg et al., 2020 https://doi.org/10.5194/bg-17-4261-2020.

Text added (line 92) is *'The elevation range is 9 to 187 m.a.s.l and the topography is mostly gently rolling slopes with some deep valleys (Marsh et al., 2010). Further details about the vegetation characteristics are available in Grünberg et al., 2020'.*

A more detailed classification of Figure 1 is shown below for discussion purposes. Slopes were generally less than 12 degrees, and although steeper slopes were present (dark green or white in the figure below), they formed a small proportion of the scene. We chose a simplified version of this figure for the manuscript in order to provide a more concise representation.

[Figure]

2) Line 132: How large is the difference between the Rayleigh-Jeans equivalent TB and the physical TB based on the Planck function (especially at the highest frequency used)? Give some typical examples. The difference between the two starts to diverge with increasing frequency and decreasing temperature.

At 89 GHz the difference is 2.1K, increasing to 5.8K at 243 GHz. This is based on simulation of the snowpits at an incidence angle of 55 deg. As discussed in lines 244-247, when we compared simulations to airborne observations we did an approximate conversion of the simulations to Rayleigh-Jeans Tb. The error of the approximation we used is less than 0.1K even for the minimum brightness temperatures observed and highest frequencies.

*(For discussion record – the manuscript has not been changed in this regard)*

3) The use of SSA in Table 2 and elsewhere: It would be easier for the reader to get the correlation length in mm (Eq. 1) than SSA in kg/m$^2$ because the wavelength is in mm, too.

We have presented the data collected in Table 2 rather than the processed data. This is to allow the reader to use the data for their own purposes. We have given the necessary information in equation 1 and in the accompanying text to allow the reader to calculate the correlation length in the same way that we have done. We do not propose to change the paper, but include the same table here with correlation length for those interested to pick this information out from this discussion.

| Pit | Date | Depth [m] | Density [kg m$^{-3}$] | | | $l_{ex}$ [mm] | | | Topographic Index |
|-----|------|-----------|------|------|------|------|------|------|------|
| | | | SS | WS | DH | SS | WS | DH | |
| A02C | 15/03/2018 | 0.2 | - | 298 | 255 | - | 0.1 | 0.44 | Valley |
| A02E | 15/03/2018 | 0.2 | - | 328 | 282 | - | 0.07 | 0.27 | Valley |
| A02W | 14/03/2018 | 0.22 | 252 | 323 | 249 | 0.08 | 0.11 | 0.3 | Valley |
| A03C1 | 17/03/2018 | 0.2 | 40 | - | 230 | 0.1 | - | 0.28 | Valley |
| A03E | 17/03/2018 | 0.42 | 159 | - | 264 | 0.06 | - | 0.37 | Valley |
| A03W | 17/03/2018 | 1.09 | 132 | 368 | 270 | 0.06 | 0.06 | 0.28 | Slope |
| A04C | 16/03/2018 | 0.31 | - | 314 | 226 | - | 0.09 | 0.33 | Plateau |
| A04C1 | 16/03/2018 | 0.27 | - | 271 | 297 | - | 0.09 | 0.35 | Plateau |
| A04N | 16/03/2018 | 0.27 | - | 302 | 272 | - | 0.14 | 0.35 | Plateau |
| A04N1 | 16/03/2018 | 0.24 | - | 232 | 265 | - | 0.07 | 0.2 | Plateau |
| A04S | 16/03/2018 | 0.38 | - | 332 | 257 | - | 0.08 | 0.28 | Plateau |
| A04S1 | 16/03/2018 | 0.4 | - | 308 | 262 | - | 0.09 | 0.29 | Plateau |
| MetS | 22/03/2018 | 0.62 | - | 297 | 252 | - | 0.06 | 0.24 | Plateau |
| A05C | 21/03/2018 | 0.3 | 96 | 380 | 246 | 0.06 | 0.08 | 0.34 | Slope |
| A05C1 | 20/03/2018 | 0.66 | 60 | 324 | 251 | 0.09 | 0.09 | 0.36 | Plateau |
| A05E | 20/03/2018 | 0.47 | 65 | 310 | 257 | 0.07 | 0.12 | 0.31 | Plateau |
| A05N | 21/03/2018 | 0.26 | 58 | 367 | 277 | 0.06 | 0.1 | 0.28 | Slope |
| A05W | 20/03/2018 | 0.94 | 75 | 336 | 202 | 0.07 | 0.07 | 0.34 | Plateau |
| A06C | 18/03/2018 | 0.24 | 158 | 310 | 244 | 0.07 | - | 0.36 | Plateau |
| A06N | 18/03/2018 | 0.19 | 52 | 222 | 216 | 0.06 | 0.05 | 0.3 | Plateau |
| A06S1 | 18/03/2018 | 0.24 | 60 | 285 | 222 | 0.08 | 0.18 | 0.42 | Plateau |
| A07C | 21/03/2018 | 0.45 | 86 | 299 | 263 | 0.06 | 0.08 | 0.33 | Slope |
| A07W | 21/03/2018 | 0.32 | 76 | 336 | 269 | 0.06 | 0.06 | 0.24 | Plateau |
| A08C | 20/03/2018 | 0.31 | 90 | 287 | 238 | 0.06 | 0.12 | 0.32 | Plateau |
| A08E | 20/03/2018 | 0.36 | 73 | 421 | 283 | 0.06 | 0.06 | 0.31 | Plateau |
| A08W | 20/03/2018 | 0.18 | 94 | 250 | 196 | 0.06 | 0.11 | 0.48 | Plateau |
| A08W1 | 20/03/2018 | 0.24 | 80 | 205 | 258 | 0.05 | 0.14 | 0.38 | Plateau |
| A09E | 20/03/2018 | 0.35 | 127 | 319 | 292 | 0.05 | 0.1 | 0.24 | Plateau |
| A09W | 20/03/2018 | 0.39 | 38 | 307 | 349 | 0.04 | 0.06 | 0.23 | Valley |

4) The identification of snow pits in Figures 2, 4, 5, 7, and Table 2 is cumbersome when changing between text parts, Tables and Figures. Please use simple numbers from 1 to 29. You still can add things like C2, such as 1-C2 for Pit 1. This is much clearer than A0C2 because all pits are now called A0...

Thank you, we have renumbered pits as suggested. We have also changed the reference to the Areas of Interest from e.g. A04 to AOI4.

5) Line 179-180: What is the thickness range of the observed ice lens? How does SMRT treat its effects? Do coherent reflections between the upper and lower boundary play a role?

The ice lens thickness ranged from 1mm to 1cm, with mean of 2mm. These are not treated explicitly in SMRT as coherent effects have not yet been implemented.

The following text has been added (line 178): *'Coherent effects of ice lenses have not been implemented in SMRT, but dielectric contrast boundary effects of ice lenses are taken into account in this study.'*

And (line 187):

*'The measured ice lens thickness ranged from 1mm to 1cm, with a mean of 2mm.'*

6) Figures 3, 8, 10: Text labels and numbers are too small.

The font size has been increased in all of these.

7) Line 204 and 210 "Atmospheric correction": This term is irritating. Tb was adapted to flight height, not corrected. Only errors can be corrected in my understanding.

We have used the alternative phrase '*adjusted*' throughout and have relabelled the section '*Adjusting for the atmosphere'* in line 212.

8) Line 232 (and elsewhere, including the abstract) "Anisotropic atmosphere": What do you mean? An atmosphere that contains anisotropic particles, such as ice crystals? Or charged particles in a magnetic field? Or do you mean anisotropic radiance? It is clear, that radiance varies with direction, even in a plane-parallel atmosphere. Therefore, the tipping-curve method has been used for a long time. But this does not mean that the atmosphere is anisotropic.

We have used the terminology 'anisotropic atmospheric radiance'.

The text in line 242 has been amended to *'The figure in Appendix A1 demonstrates how atmospheric downwelling varies with viewing angle and therefore why it is important to represent the anisotropy of the atmospheric radiance.'*

9) Line 240: "nadir ground-based TBs". See General comments, above. Measurements may be distorted (mostly enhanced) by the effects mentioned there.

The following text has been added (line 257): '*This is quantified in terms of a difference rather than error as measurements themselves may be subject to small distortions due to shadowing of the sky and emission from the radiometers.'*

10) Figure 7: I do not understand the box symbols. The key is incomplete.

Captions now have the following additional text to aid interpretation:

Fig 7: *"Airborne data: box (interquartile range), horizontal black line (median), vertical black lines (whiskers extending from the end of each box to 1.5 times the interquartile range), black circles (outliers beyond this range); Ground data: filled orange circle (mean), vertical orange line (range); SMRT simulations: filled blue hexagon (TB using mean measured snow properties), vertical blue line (TB range using combinations of maximum and minimum measured SSA and density)."*

*Fig 8 and Fig A2: "Airborne data: box (interquartile range), horizontal orange line (median), vertical black lines (whiskers extending from the end of each box to 1.5 times the interquartile range); SMRT simulations: filled blue hexagon (TB using mean measured snow properties), vertical blue line (TB range using combinations of maximum and minimum measured SSA and density)."*

11) Figure 8: Why not without atmosphere, or with a time-constant atmosphere. It is not clear if the changes are due to the atmosphere or due to the surface.

There are minimal differences in the simulations: the same pit information is used in both sets of simulations. The only difference is the interpolation of temperature and the atmospheric emission. Within Figure 8, there are only noticeable changes in observations. The observed changes between flights can be represented by adding / removing the low density surface snow in the simulations i.e. by comparing Figure 8 and Figure A2.

*(For discussion record – the manuscript has not been changed in this regard)*

12) Figure 9 is very helpful because it shows the weather history. Its information could be used to better interpret the radiometer data. The temperature remained below freezing. No changes are expected for the ice lenses. Temperature-gradient metamorphism with slow changes only. The figure also indicates that time series of radiometric measurements at the same temporal resolution might be valuable.

We agree that a time series of radiometric observations would be hugely valuable – regretfully these do not exist for this dataset. We did originally attempt to model changes in microstructure due to temperature-gradient metamorphism between flights and found these to be negligible.

The text (line 329) has been amended to '*No significant changes are expected in layer microstructure throughout the course of the field campaign as the temperature remained below freezing and only small changes in SSA can be expected over the days between flights. However, after flight C087 on 16th March there were several snowfall events.'*

13) Table 4: Text unclear. I don't see any "effect of thin...". I only see numbers. Please clarify. Why are they all negative?

The numbers are negative because including surface snow lowers the brightness temperature. The following text has been added to the caption: '*Negative values indicate that inclusion of low-density surface snow reduces the brightness temperature'*.

14) Line 330: "This suggests that emission from the atmosphere may dominate ... at 183 GHz". It appears to me that the author did not check the actual brightness temperatures & opacities involved.

The text (line 346) has been changed to read: '*This suggests that emission from the atmosphere itself may dominate over the impact of the additional surface snow layer at 183 GHz, which is consistent with the higher measured and simulated emission at 183 GHz shown in Figure A1.'*

15) Figure 10 is very helpful. It is the only part where we clearly see the influence of the atmosphere. However, the analysis and description should be improved, e.g. on Line 333: "the atmosphere reduces the RMSE of the base simulation medians". I cannot see any RMSE in this figure. Do you mean the widths of the distributions shown? Please don't call this an error. And certainly not of the

medians. Later, on Line 338, you mention something with respect to the distributions. I was unable do understand this text.

We have changed the text so that we now use Root Mean Square Difference rather than RMSE throughout the manuscript.

We have changed this sentence (line 357) referring to distributions, so it is now more explicit which distributions are being compared: '*However, Kolmogorov-Smirnov 2-sample tests of distribution equivalence show that simulated distributions (either with or without atmosphere) are statistically different to distributions of airborne observations at a 5% significance level.*'

16) Line 345: Upper frequency limit of IBA: There is no fixed limit. The error of the approximation just increases with increasing frequency (and is larger for scattering in backward than in the forward hemisphere). "radius" should be defined, or else replaced by "correlation length".

The text (line 363) has been replaced with: '*With an estimated limit of wavenumber k_0 ~ 1.5 x radius of spheres to keep the error of the approximation within reasonable limits, as specified by Picard et al., 2022, the IBA upper frequency for ....*'

17) Line 350: "Underlying topography": Do you really mean topography, here? Or dielectric properties of the underlying ground? The topography, in terms of slope steepness, orientation, and the solid angle of open sky above the surface point is relevant at all frequencies. The discussion that follows seems vague and not enough specific to the situations of the study.

The topography of snow surface would be relevant at all frequencies, but the accumulation of snow in depressions will mean the topography of the snow surface is smoothed compared with the surface underneath the snow. Both simulations and data show less differentiation between topography classifications at higher frequencies. There is no difference in dielectric properties of the underlying ground in the simulations, so from that perspective only the measured snowpack properties (driven in part by topography) changes between pits.

The paragraph (line 370) has been amended to read:

'*Underlying topography is relevant at 89 GHz but becomes less relevant at higher frequencies. As the frequency increases, the penetration depth reduces and the sensor may only see the upper portion of snowpack. This is the dominant effect and results in smaller differentiation between TB classified by ground topography. However, structural changes and spatial variability in snowpack properties driven by topography may result in a topographical signal in the TB despite the signal not penetrating to the base of the snowpack. Small differences between topographical types persist even at 243 GHz in Figure 8.*'

18) The discussion on Lines 366 to 370 indicates that the selection of sensors used was not optimal. A mapping sensor with sufficient spatial resolution would have been more helpful, even if it is at a single frequency, such as 89 GHz. As an alternative a movable radiometer on a sledge would also give information on the spatial variability. Of course, this is no argument against the mentioned snow micropenetrometer. Both together would be excellent.

Rather than non-optimal selection of sensors, this discusses a limitation of the study. The sensors used are used or planned atmospheric frequencies and the question is whether we can use electromagnetic modelling to account for the snow emission and ultimately retrieve atmospheric information from these data. There will always be challenges in comparing point and areal

measurements, and ideally we would have ground-based instruments at these frequencies too, used to make spatially distributed observations. We hope that such a field campaign may be possible in the future.

*(For discussion record – the manuscript has not been changed in this regard)*

19) Finally, I am surprised about the large standard deviation of all simulated TB values. What is the reason? And how can you get more specific results that better focus on the actual situations?

The large standard deviation comes from the range of measured SSA and density within individual snowpits. They cover a plausible range of observations, with the 'base case' giving the best estimate. In order to get more specific results, we would need the 3D structure of the snow over the footprint of the sensor and solution of Maxwell's equations e.g. with NMM3D (e.g. Xu et al., 2010: https://doi.org/10.1109/JSTARS.2010.2053919), but we do not have this information.

*(For discussion record – the manuscript has not been changed in this regard)*

Reviewer 2- Alan Geer

Possibly for the first time, this paper demonstrates good agreement between snow radiative transfer simulations (driven by snow pit measurements) and downlooking microwave observations at higher frequencies, i.e. 89 GHz to 243 GHz. This illustrates a path towards using snow radiative transfer, driven by multi-layer snow models, in the assimilation of microwave observations for both weather forecasting and for the inference of surface snow properties. The coupling of ARTS and SMRT models (and the demonstration of why this is important) is also an important step. The paper also illustrates some of the remaining difficulties to be solved. One of these is the occasionally large mismatches between point snow brightness temperature (TB) measurements and airborne TB measurements with fields of view up to 100m. Another is the drop in TB of up to around 20 K observed from one flight to the next, which was attributed to fresh snow on the surface, and illustrates strong temporal variability.

Overall, this paper is an important step forward, it will be of great interest to many scientists in the field, and it is well presented. However, there are a few areas where the methodology or results could be better explained, there are some possible uncertainties that might deserve more consideration, and it is important that the abstract and conclusions should clearly indicate the scope and limitations of the work.

We thank Dr. Geer for the positive overview of this paper, which brings out the scope and limitations of the work more clearly.

Main points

1) As described in the paper's abstract, coupling ARTS and SMRT is a major developmental step. However, for such an important part of the paper there is very little detail. For example it is not clear how the downwelling atmospheric radiation field is represented by ARTS and then coupled into SMRT (presumably as radiances at the quadrature angles of the discrete ordinates solver used in SMRT, but this is not stated). Assuming ARTS and SMRT are not "fully" coupled, by which I mean a discrete ordinates problem is solved simultaneously in the snow and atmosphere, I imagine that ARTS is called first to simulate the downwelling radiance field, the upwelling radiance at the observation angle, and the surface-to-aircraft transmittance. Then presumably SMRT is called and its output corrected to aircraft level with the paper's equation 2. These issues should be clearly discussed in the paper in section 2.4. It would also be good to have details of the ARTS radiative transfer solver method, mainly just to exclude the unlikely scenario that atmospheric scattering is being represented too (which could need "full" coupling of the solvers).

We appreciate and agree this needs further detail. It is correct that the two are not fully coupled and a flowchart has been included as Figure A2 with the following caption: *Flowchart demonstrating SMRT-ARTS coupling and processing steps for comparing with observations. ARTS atmospheric properties are used to calculate upwelling TB (TBup), downwelling TB (TBdown) and atmospheric transmissivity matrices used in the SMRT snowpacks. Although ARTS is configured as an initial step under the assumption of a surface blackbody, calculation of the atmospheric properties is carried out dynamically during the SMRT simulation as the matrices depend on the array of incidence angles and frequencies of the sensors. Equation \ref{eqn:adjust_tb} is used to adjust SMRT simulations to surface height for comparison with ground-based sensors (Figure 4), and for adjustment of ground observations to account for the atmosphere beneath the aircraft (Figures 5 and 7).*

The following text has been added to Section 2.4 (line 223):

*A flowchart illustrating the loose coupling between SMRT and ARTS and processing steps is given in Appendix A2.*

*The following text has been added (line 214) to include more details on the ARTS configuration: 'The ARTS Clear Sky (non-scattering) solver is used for a 1D atmosphere. The sensor is represented using a "top-hat" channel response in each of the two sidebands, with a frequency resolution of 0.1GHz.'*

2) Some of the descriptions of how Arctic microwave observations are used at NWP centres (with ECMWF as the main example) could be made more precise. Microwave sounding radiances are used over snow and sea-ice surfaces if the surface contribution is small enough. For example the 183+/-3 GHz channels are usually assimilated over sea-ice and snow, whereas 183+/-7 GHz channels are not. Also, one of the main problems with ice and snow surfaces, from an NWP point of view, is that a constant surface emissivity cannot be assumed. Over non-snow land surfaces, the dynamic emissivity retrieval technique typically assumes that an emissivity retrieval can be extrapolated using a constant in frequency approximation (for example an 89 GHz retrieval is used as the surface emissivity for 183 GHz assimilation over non-snow surfaces). A few more detailed points illustrating these issues:

line 28-29: "data over Arctic regions" could more precisely be "surface-sensitive data over Arctic regions" and the reason for the data exclusion is usually the possible presence of snow and ice.

Text (line 28) now reads: *"However, surface-sensitive data over Arctic regions are frequently discarded because of the difficulty in accounting for the surface component..."*

And in the subsequent sentence, it now reads: *"Previous research has indicated potential benefits of the assimilation of surface-sensitive microwave data over Arctic regions, and that forecast improvements may extend to lower latitudes in the medium-range, with some uncertainty in mechanisms and magnitude..."*

line 30: "potential benefits of .. microwave data over Arctic regions" - but some Arctic microwave data is already being assimilated operationally, particularly in summer, as illustrated in the Lawrence et al. (2019) studies, and as is described clearly on lines 35-38.

We have removed remove the word 'potential', used as 'potential benefits' twice in this paragraph, to align with current operational procedures.

line 46: Baordo and Geer (2016) describe the assimilation of only snow-free land surface data, for the SSMIS instrument, and they eliminated surface-sensitive observations at latitudes greater than 60 degrees or for surface temperatures less than 278 K. Hence the point about using atlas in these possible-snow areas is not so relevant. Within Geer et al. (2014) there is a description of subsequent work that extended SSMIS usage over snow and sea ice surfaces following the above-described template. This actively assimilates 183+/-3 GHz and higher peaking channels. The dynamic emissivity retrieval is made at 150 GHz and then assumed to be valid also at 183 GHz, making sure the extrapolation in frequency is as small as possible (but even this small extrapolation induces errors that are too large to permit assimilation of channels that have stronger surface sensitivity, like 183+/-7 GHz). This snow and ice dynamical emissivity retrieval approach started at ECMWF even earlier with clear-sky MHS assimilation following the work of Di Tomaso et al. (2013: Assimilation of ATOVS radiances at ECMWF: third year EUMETSAT fellowship report. EUMETSAT/ECMWF Fellowship Programme Research Report No. 29, available from http://www.ecmwf.int.)

The text (line 46) has been amended to read:

*A dynamic emissivity retrieval was proposed by Di Tomaso (2013) and Geer et al. (2014), where land surface emissivities derived at 90 GHz were used at 183 +/- 3 GHz and higher frequencies. However, this is is not applicable for channels with high surface sensitivity e.g. 183 +/- 7 GHz as the errors are too large.*

(The reference states 90 GHz for snow-covered land surfaces whereas 150 GHz is used for snow-covered sea ice).

line 48: "microwave emissivity is highly spatially variable" - this could be a place to mention that it is also highly variable in frequency.

This sentence (line 49) now reads: *"Particularly over snow, the microwave emissivity is highly spatially variable, highly dependent on frequency and has high uncertainty due to its sensitivity to the microstructure (grain size, shape and spatial arrangement at the micrometer scale) of the snow."*

Just a discussion point, but these dynamic surface approaches are continuing to be improved for NWP, and in particular we are starting to improve representations of the frequency dependence of surface emissivity. Compared to the more physical approach of the paper under review, the dynamic approach has the advantage of being able to adapt the surface to match what is in the sensor's the field of view, thus dealing with the time and space heterogeneity issues that are well illustrated in the paper, and hence they may continue to provide strong competition for the fully physical approach for some time to come.

This is a very welcome discussion point and incoming improvements to NWP. We hope that the physical approaches will underpin the representations of frequency dependence of surface emissivity and will help drive improvements in the land surface model representation also.

The following text has been added to the discussion (line 429):

*Alternative approaches with dynamic emissivity depending on frequency can be supported through SMRT modelling.*

3) It would have been good to discuss the surface characteristics of the Trail Valley Creek site and how they relate to possible uncertainties in the surface radiative transfer. In particular, vegetation, since it appears the surface is being modelled as bare soil. The satellite pictures seem to show trees in the valleys and the possibility of grass or shrubs on the plateaus. Could vegetation have impact on the radiative transfer, particularly if it contains some liquid water, and particularly as the snow cover is not deep, e.g. 20 - 100 cm (lines 109-110)?

This is a very good point and could certainly impact the quality of the simulations. It's worth noting that the dominant land surface is tussocks (37%) followed by dwarf shrubs (24%), whereas trees only constitute 2% Grünberg et al., 2020 https://doi.org/10.5194/bg-17-4261-2020.

The following text has been added to the site description (section 2.1, line 93):

*The dominant land surface is tussocks (37%) followed by dwarf shrubs (24%), whereas trees only constitute 2%. Further details about the vegetation characteristics are available in Grüneburg et al., 2020.*

The following table has been added to the appendix, which gives information on vegetation within individual pits

| Pit | Vegetation notes |
|---|---|
| 1-2C | Tussocks and a few shrub twigs |
| 2-2E | Tussocks and dwarf shrubs |
| 3-2W | Grass tussocks |
| 4-3C1 | Grass (very loose snow towards bottom, blocked by vegetation) |
| 5-3E | Lots of shrubs to 60cm |
| 6-3W | - |
| 7-4C | Tussocks and twigs |
| 8-4C1 | Tufts of grass |
| 9-4N | Tussocks and twigs |
| 10-4N1 | - |
| 11-4S | Tussocks and twigs |
| 12-4S1 | Lichen |
| 13-MetS | - |
| 14-5C | - |
| 15-5C1 | Lichen. Trees around pit |
| 16-5E | Further from the trees than the other. Lichen |
| 17-5N | - |
| 18-5W | Lichen, shrubs, trees around snowpit |
| 19-6C | Shrub, lichen, vegetation 7cm tall in pit |
| 20-6N | Grass and lichen |
| 21-6S1 | Lichen, small bushes |
| 22-7C | 2m shrub in area, 30cm shrub in pit |
| 23-7W | - |
| 24-8C | Lichen |
| 25-8E | - |
| 26-8W | - |
| 27-8W1 | Grass and moss |
| 28-9E | - |
| 29-9W | - |

Ideally we would have a radiative transfer model that simulates the effects of the vegetation, but this is not yet possible with SMRT. Vegetation was noted in many but not all pits, as shown in the Table above (new Table A1). Emission from the vegetation not accounted for could potentially contribute to an underestimation in simulated brightness temperature. However, the contributions from twigs and grasses are likely to be small. The change in snow structure due to vegetation in pit 4-3C1 (A03C1) – 'very loose snow towards bottom, blocked by vegetation' could be a contributing factor in the discrepancy between observations and simulations.

The following text has been added to the discussion (line 411):

*Vegetation may have contributed to modelling error as this was noted in many pits (see Table A1) but was not taken into account in modelling as this has yet to be implemented in SMRT. However in pits with lots of vegetation noted (shrubs to 60cm in 5-3E and 30cm shrub in pit 22-7C), SMRT base simulations are within 1.5 times the interquartile range of airborne observations, showing these effects may be small. Although contributions from twigs and grasses are likely to be small, the change in snow structure due to vegetation in pit 4-3C1 (very loose snow towards bottom, blocked by vegetation) could be a contributing factor in the discrepancy between observations and simulations. It is difficult to sample snow density and SSA within vegetation, and shrubs alter the snowpack properties, increasing depth hoar (Royer et al., 2021).*

4) There could be some more investigation of the way the temperature profile is determined, and whether this has any bearing on the radiative transfer uncertainties. Lines 168-170 describe a linear extrapolation from the air temperature (ultimately from dropsondes?) through the snowpack to a stable lower layer temperature. Is this sufficient to represent the relative complex dependence of the snow temperature profile on the surface air temperature, particularly its insulation properties and speed of heat transfer? For example, when trying to explain the drop in brightness temperature between flights C087 and C090, could this be relevant? Looking at Figure 9, at the time of the C090 flight, could the snow still be cold after a night that dropped below -25 degrees C, and hence has not yet caught up with the rapid rise in the air temperature?

All flights were around 2pm, so residual cold in upper snowpack layers from nocturnal cooling is unlikely to have persisted. At these frequencies the emission will be more sensitive to surface snow layer temperatures and less sensitive to internal snowpack temperatures. The interpolation between air temperature and basal temperature is taken from the ground station measurement, not the dropsondes (i.e. from met data in Figure 9), and we are assuming the air temperature value measured at the Met Station is representative over the whole TVC site. This is a simplification, and a better method would be e.g. snowpack modelling. However, there were only minor differences between using the measured pit temperatures (at the time of pit measurements) and interpolated temperature estimates (at the time of flights) – see figure below, so full snowpack temperature profile modelling was not deemed appropriate.

The figure below is from an earlier draft of the paper that used measured pit temperatures rather than interpolated from the air temperature. Although other amendments were also made (e.g. inclusion of missing ice lens information in two pits), a comparison between the figure below and Figure 7 in the paper illustrates that the effects of temperature representation are minor.

[Figure]

*(For discussion record – the manuscript has not been changed in this regard)*

5) It could be worth specifying also the type of seasonal snow in the abstract and conclusions. Currently on line 406 the work is described as relating to "an Arctic tundra snow environment" but that could more precisely be "an Arctic tundra snow environment in late winter". In order to use satellite observations for weather forecasting globally and in all seasons over snow and sea-ice, we will need to be able to simulate many other snow types such as wet snow and including diurnal cycles of freeze and thaw.

This is a good point and we have changed this sentence (line 440), so it now reads: "*In this study SMRT was evaluated at frequencies between 89 and 243GHz in an Arctic tundra snow environment with dry snowpacks, with the atmospheric contribution estimated with ARTS.*"

This increases the temporal range of applicability of this study using the specificity of snowpack conditions without restricting to late winter, which could be misconstrued as the snowmelt period.

The following text has also been added to the discussion to highlight the issue of wet snow (line 431):

*This modelling study encompassed dry snow conditions only, but wet snow conditions must also be considered in future work for operational numerical weather prediction models. Although the emissivity and temperature of uniformly wet snow are well-known, within-footprint spatial distribution of melt is important for simulation of brightness temperature (e.g. Vuyovich et al., 2017). The ability of land surface models to capture spatial and temporal variability in wet snow, especially freeze-thaw cycles, is important if these data are to be used to their full capacity in numerical weather prediction.*

 Minor points

line 40 - 19, 37 and 89 GHz channels are extensively used for water vapour, cloud and precipitation assimilation, but the statement that "window frequencies around 19, 37 and 89 GHz are used to obtain information about the surface (e.g. snow)" could be misread to exclude this and to imply that these frequencies are not useful for the atmosphere.

Sentence (line 40) now reads: "*Atmospheric window frequencies around 19, 37 and 89 GHz are typically chosen for applications requiring information about the surface (e.g. snow) as they are less sensitive to the atmosphere.*"

line 44-45 - "forecast and analysis"? rather than "forecast analysis" which is confusing.

Change made.

line 115 - if the sled measurements are nadir to a snow surface that may be sloping, is any adjustment made when sled measurements are mapped to true nadir aircraft measurements?

No adjustments were needed – sled measurements were made over near horizontal surfaces. Text (line 119) has been updated so it now reads: "*The radiometer was mounted on a sled at a height of approximately 1.5 m above, and at an angle nadir to near horizontal snow surfaces.*"

line 196 - "representing the layer density and SSA by the largest and smallest observed values" - it's not clear whether this means within a single pit, or across all pits.

Sentence (line 203) now reads: *"However, the largest impact on TB was found by representing the layer density and SSA by the largest and smallest observed values within each layer of each pit."*

line 197 - the "full range of plateau airborne observations" deserves some explanation, as at this stage it's really not clear that (presumably) this means across the two flights and incorporating all plateau measurements in the relevant area illustrated in figure 6 and following the comparison strategy described in lines 262-265. It might be worth re-ordering some of this information (e.g. to put it in the section on aircraft data?)

We agree and think this can be addressed by adding greater specificity to the sentence, rather than re-ordering between sections. Sentence (line 204) now reads: *"Over areas within AOI4 classified as plateau, Including all effects resulted in a TB range of 164-193 K, close to the full range of airborne observations from the C087 flight (163-201 K in Table 3)."*

line 221-227 - in the adjustment of the background atmospheric profile to fit aircraft-measured downwelling radiances, can the dropsonde profile below the aircraft be modified to fit observations? It's not clearly excluded in the text. And how representative is the lowermost dropsonde air temperature of the snow temperature? (See main point 4)

The dropsonde temperatures were not used in the interpolation of snow temperatures: this information comes from the meteorological station. This is because no downwelling observations were available below the aircraft, it is not possible to modify the profile below aircraft height. We have updated the text (line 231) to read. *"Temperature and water vapour profiles used as input for ARTS were retrieved for each AOI in each flight. Background profiles were taken from a combination of dropsonde profiles, from sondes released before the low-level AOI runs, and profiles from the Met Office operational global NWP model (above sonde height). The retrieval adjusts these background profiles to match aircraft-level downwelling observations in the vicinity of each AOI at 183±1, ±3 and ±7 GHz. Because downwelling observations are only available above the aircraft, the profile below aircraft height (~590 m in the AOIs) is not adjusted in the retrieval."*

For additional information, the sondes were dropped when the aircraft was at a higher altitude before the AOI runs (~7500 m in C087, and ~1800 m in C090), meaning the sonde profiles start above the altitude at which downwelling observations were made in the AOIs (~590 m). A portion of the dropsonde profile is therefore modified during the retrieval, just nothing below the height of the aircraft in the AOIs. The section of profile below the aircraft is relatively small compared to the total atmospheric profile (max altitude 80,000 m), therefore any uncertainty resulting from this is also expected to be relatively small.

Figure 4 caption (figure 5 similarly) - the significance of the square could be explained in words in the caption (the lines indicating that it is a zoom are faint and easy to miss), The caption should also explain the meaning of the error bars

The zoom box has been made more obvious and the caption now reads:

Fig 4: *"Comparison between SMRT simulations and ground-based radiometer observations at 89 GHz, nadir. Blue circles (mean of ground observations and TB simulation using mean measured snow properties), blue lines (range of ground observations and TB simulation range using combinations of*

*maximum and minimum measured SSA and density). The zoom box is used to provide space to label these specific pits."*

Fig 5: *"Comparison between ground-based observations of brightness temperature and airborne brightness temperature at 89 GHz for pits where observations were available. Airborne observations from both C087 and C090 flights were used, and ground-based observations have been adjusted to height of aircraft. Blue circles (mean of ground observations and median of airborne observations), blue lines (range of ground observations and inter-quartile range of airborne observations). The zoom box is used to provide space to label all pits. Note that pits 7-12 are paired pits within close proximity."*

line 241-242 - linked to Figure 4 and the need to state clearly what the error bars mean, it's not clear how the "range of simulations" mentioned here is being generated.

We have improved clarity in the caption for Figure 4 (see previous comment), which now explicitly describes what each bar means around the symbols represent. So, the 'error bars' are now explicitly referred to as 'simulation range' in the caption, which better corresponds directly to the current use of 'range of of simulations' in the highlighted paragraph.

Figure 7 and 8 captions - need a careful description of the meaning of the various boxes, whiskers and spots.

Captions now have the following additional text to aid interpretation:

Fig 7: *"Airborne data: box (interquartile range), horizontal black line (median), vertical black lines (whiskers extending from the end of each box to 1.5 times the interquartile range), black circles (outliers beyond this range); Ground data: filled orange circle (mean), vertical orange line (range); SMRT simulations: filled blue circle (TB using mean measured snow properties), vertical blue line (TB range using combinations of maximum and minimum measured SSA and density)."*

Fig 8 and Fig A2: *"Airborne data: box (interquartile range), horizontal orange line (median), vertical black lines (whiskers extending from the end of each box to 1.5 times the interquartile range); SMRT simulations: filled blue circle (TB using mean measured snow properties), vertical blue line (TB range using combinations of maximum and minimum measured SSA and density)."*

Line 383 - this RMSE calculation is a headline result from the paper, quoted in the abstract, so it should be clear how it is obtained. For me the description "RMSE of the base simulation medians by frequency and flight" is not quite clear enough. For example whether this really is the RMS of the SMRT base simulation median minus the observation median and, if I understand correctly, that means the sample over which the RMS is computed is of size ten, e.g. "across 5 frequencies and 2 flights"? Could this be more precisely described in the abstract too, noting specifically the use of medians in the calculation? Because if it is based on medians, then we might expect the RMSE comparing the errors of individual pits to individual surface categories and AOIs to be somewhat higher. That would also be a useful figure to calculate.

We have now used Root Mean Square Difference rather than RMSE to describe this. This is the difference between medians so it is correct that n=10. This has been clarified in the abstract.

The abstract text (line 12) has been amended to read: *Medians of simulations compared well with medians of observations, with a root mean squared error of 14 K, across five frequencies and two flights (n=10).*

For individual surface categories within AOIs against pits, the RMSD is 35.7 K excluding the atmosphere and 18.4K with the atmosphere included for flight C087 (n=145). For flight C090 the RMSD without atmosphere is 29.2K and with the atmosphere is 21.7K. These have been included in the revised manuscript, but will not be presented as a combined figure for both flights because of the difference for C090 likely caused by the thin low density snow layer.

Text in the results section (line 348) now reads: *The importance of including the atmosphere at different frequencies is demonstrated in Figure 10. Overall, inclusion of the atmosphere reduces the root mean squared difference (RMSD) of the base simulation medians by frequency and flight from 23 K to 14 K. At an individual pit level, comparison with airborne data of the same topography classification (i.e. plateau, slopes or valleys) reveals that inclusion of the atmosphere reduces the RMSD from 35.7 K to 18.4K for flight C087 (n=145). For flight C090 the RMSD without atmosphere is 29.2K and with the atmosphere is 21.7K.*

Line 396-397 - main point 2 again: "In current numerical weather prediction models, microwave emissivity is assumed to be constant over snow-covered surfaces or derived from a monthly climatology, with errors too large to be able to use satellite observations in the Arctic": Dynamic emissivity retrievals have been used over snow and sea-ice at ECMWF to allow assimilation of 183+/-3 GHz channels (and higher peaking channels) since the work described in Di Tomaso et al. (2013) and Geer et al. (2014). Hence the snow emissivity does not usually come from atlas and it is not assumed constant in time or space (but it is assumed constant with frequency from 150 to 183 GHz). Nonetheless, the dynamic emissivity retrievals are not yet good enough to permit assimilation of strongly surface sensitive channels (e.g. 183+/-7 GHz) over snow so there still is plenty that can be improved by physical modelling as described in the paper under review.

This is a valuable discussion point – dynamic emissivity retrievals were briefly discussed earlier in the paper with revised text (see response to comment about line 45). In the discussion we have amended the text (line 421) to read:

*In current numerical weather prediction models, microwave emissivity is assumed to be constant over snow-covered surfaces, is derived from a monthly climatology or is retrieved dynamically with emissivity assumed constant over frequency (Di Tomaso, 2013 and Geer et al., 2014). However, in some channels, errors in these approaches are too large to be able to use satellite observations in the Arctic.*

Line 422-423 - "the addition of fresh, low density precipitation and a later wind event that removed it over the space of a few days caused differences in observed brightness temperatures." - is the attribution of these changes in TB to the fresh snow event certain enough to be able to say for definite it "caused" it here in the conclusion, rather than to say "likely caused", for example?

While is it consistent with the simulations and could not otherwise be explained, we accept that it is not possible to attribute this conclusively. Line 456 now reads *"i.e. the addition of fresh, low density precipitation and a later wind event that removed it over the space of a few days likely caused differences in observed brightness temperatures."*

---

## Author Response (AR2)

Dear Patricia – thank you for your positive response to our manuscript changes. Please find below a summary of the further changes we have made in response to the reviewer comment. This reviewer comment is shown in black, our response in blue, and the *changed text in the manuscript in blue italics*. Line number 46 refers to the new version of the manuscript.

Many thanks on behalf of all authors,

Mel

Just a few suggestions on the new text (I thought) describing the dynamic emissivity retrieval over snow and ice and the usage at ECMWF, which has not come across perfectly and I would suggest a slight rewrite. Ideally what should be addressed in this sentence is the approach where, for ice and snow surfaces only, we do the dynamic retrieval in the window channel at 150-166 GHz and apply this emissivity, without modification, to the 183 GHz channels. Another citation to consider specifically for dynamic emissivity retrievals over snow and ice surfaces as a replacement for, or addition to, the ECMWF citations would be prior work at Meteo France (who used 89 GHz for the retrieval and a frequency-based adjustment to the retrieved emissivity to make it valid at 183 GHz) - https://doi.org/10.1175/2009MWR3071.1

If I was rewriting this sentence my suggestions would be (a) indicate that it specifically focuses on snow and ice surfaces, as a progression beyond the work of Baordo and Geer on non-snow surfaces that is mentioned in the previous sentence; (b) say that the relevant emissivity retrieval channel over snow and ice surfaces is 157 GHz (if talking about MHS, as was the case in the citations being made) rather than saying 90 GHz (90 GHz is the channel used for dynamic emissivity retrievals at ECMWF for nonsnow land surfaces, to be used for 183 GHz channels); (c) for fairness consider acknowledging some of the prior Meteo France work in this area, e.g. "... following earlier work by Bouchard et al. (2010)."

We thank Dr. Geer for this clarification. We have amended the text at line 46 to read:

*A dynamic emissivity retrieval was proposed by Di Tomaso et al. (2013) and Geer et al. (2014), where land surface emissivities derived at 90 GHz were used at 183 ± 3 GHz and higher frequencies over snow-free land. However, this is not applicable for channels with high surface sensitivity e.g. 183 ± 7 GHz as the errors are too large. Following the earlier work of Bouchard et al. (2010), the relevant window channel to derive emissivity for snow- and ice-covered surfaces is 157 GHz, which is used without modification at 183 GHz.*

Please note that Competing Interests has now been amended to
*'At least one of the (co-)authors is a member of the editorial board of The Cryosphere.'*
As this is not in the main body of the text, this has not been picked up by the pdf difference pipeline and is not highlighted with different colour text in the Tracked Changes version.